# Doubly Robust Self-Training

**Banghua Zhu**
Department of EECS
UC Berkeley
banghua@berkeley.edu

**Mingyu Ding**
Department of EECS
UC Berkeley
myding@berkeley.edu

**Philip Jacobson**
Department of EECS
UC Berkeley
philip_jacobson@berkeley.edu

**Ming Wu**
Department of EECS
UC Berkeley
mingwu@berkeley.edu

**Wei Zhan**
Department of EECS
UC Berkeley
wzhan@berkeley.edu

**Michael I. Jordan**
Department of EECS
UC Berkeley
jordan@berkeley.edu

**Jiantao Jiao**
Department of EECS
UC Berkeley
jiantao@berkeley.edu

## Abstract

Self-training is an important technique for solving semi-supervised learning problems. It leverages unlabeled data by generating pseudo-labels and combining them with a limited labeled dataset for training. The effectiveness of self-training heavily relies on the accuracy of these pseudo-labels. In this paper, we introduce doubly robust self-training, a novel semi-supervised algorithm that provably balances between two extremes. When the pseudo-labels are entirely incorrect, our method reduces to a training process solely using labeled data. Conversely, when the pseudo-labels are completely accurate, our method transforms into a training process utilizing all pseudo-labeled data and labeled data, thus increasing the effective sample size. Through empirical evaluations on both the ImageNet dataset for image classification and the nuScenes autonomous driving dataset for 3D object detection, we demonstrate the superiority of the doubly robust loss over the standard self-training baseline.

## 1 Introduction

Semi-supervised learning considers the problem of learning based on a small labeled dataset together with a large unlabeled dataset. This general framework plays an important role in many problems in machine learning, including model fine-tuning, model distillation, self-training, transfer learning and continual learning (Zhu, 2005; Pan and Yang, 2010; Weiss et al., 2016; Gou et al., 2021; De Lange et al., 2021). Many of these problems also involve some form of distribute shift, and accordingly, to best utilize the unlabeled data, an additional assumption is that one has access to a teacher model obtained from prior training. It is important to study the relationships among the datasets and the teacher model. In this paper, we ask the following question:

> *Given a teacher model, a large unlabeled dataset and a small labeled dataset, how can we design a principled learning process that ensures consistent and sample-efficient learning of the true model?*

37th Conference on Neural Information Processing Systems (NeurIPS 2023).

Self-training is one widely adopted and popular approach in computer vision and autonomous driving for leveraging information from all three components (Lee, 2013; Berthelot et al., 2019b,a; Sohn et al., 2020a; Xie et al., 2020; Jiang et al., 2022; Qi et al., 2021). This approach involves using a teacher model to generate pseudo-labels for all unlabeled data, and then training a new model on a mixture of both pseudo-labeled and labeled data. However, this method can lead to overreliance on the teacher model and can miss important information provided by the labeled data. As a consequence, the self-training approach becomes highly sensitive to the accuracy of the teacher model. Our study demonstrates that even in the simplest scenario of mean estimation, this method can yield significant failures when the teacher model lacks accuracy.

To overcome this issue, we propose an alternative method that is *doubly robust*—when the covariate distribution of the unlabeled dataset and the labeled dataset matches, the estimator is always consistent no matter whether the teacher model is accurate or not. On the other hand, when the teacher model is an accurate predictor, the estimator makes full use of the pseudo-labeled dataset and greatly increases the effective sample size. The idea is inspired by and directly related to missing-data inference and causal inference (Rubin, 1976; Kang and Schafer, 2007; Birhanu et al., 2011; Ding and Li, 2018), to semiparametric mean estimation (Zhang et al., 2019), and to recent work on prediction-powered inference (Angelopoulos et al., 2023).

## 1.1 Main results

The proposed algorithm is based on a simple modification of the standard loss for self-training. Assume that we are given a set of unlabeled samples, $\mathcal{D}_1 = \{X_1, X_2, \cdots, X_m\}$, drawn from a fixed distribution $\mathbb{P}_X$, a set of labeled samples $\mathcal{D}_2 = \{(X_{m+1}, Y_{m+1}), (X_{m+2}, Y_{m+2}), \cdots, (X_{m+n}, Y_{m+n})\}$, drawn from some joint distribution $\mathbb{P}_X \times \mathbb{P}_{Y|X}$, and a teacher model $\hat{f}$. Let $\ell_\theta(x, y)$ be a pre-specified loss function that characterizes the prediction error of the estimator with parameter $\theta$ on the given sample $(X, Y)$. Traditional self-training aims at minimizing the combined loss for both labeled and unlabeled samples, where the pseudo-labels for unlabeled samples are generated using $\hat{f}$:

$$\mathcal{L}^{\mathsf{SL}}_{\mathcal{D}_1, \mathcal{D}_2}(\theta) = \frac{1}{m+n} \left( \sum_{i=1}^{m} \ell_\theta(X_i, \hat{f}(X_i)) + \sum_{i=m+1}^{m+n} \ell_\theta(X_i, Y_i) \right).$$

Note that this can also be viewed as first using $\hat{f}$ to predict all the data, and then replacing the originally labeled points with the known labels:

$$\mathcal{L}^{\mathsf{SL}}_{\mathcal{D}_1, \mathcal{D}_2}(\theta) = \frac{1}{m+n} \sum_{i=1}^{m+n} \ell_\theta(X_i, \hat{f}(X_i)) - \frac{1}{m+n} \sum_{i=m+1}^{m+n} \ell_\theta(X_i, \hat{f}(X_i)) + \frac{1}{m+n} \sum_{i=m+1}^{m+n} \ell_\theta(X_i, Y_i).$$

Our proposed doubly robust loss instead replaces the coefficient $1/(m+n)$ with $1/n$ in the last two terms:

$$\mathcal{L}^{\mathsf{DR}}_{\mathcal{D}_1, \mathcal{D}_2}(\theta) = \frac{1}{m+n} \sum_{i=1}^{m+n} \ell_\theta(X_i, \hat{f}(X_i)) - \frac{1}{n} \sum_{i=m+1}^{m+n} \ell_\theta(X_i, \hat{f}(X_i)) + \frac{1}{n} \sum_{i=m+1}^{m+n} \ell_\theta(X_i, Y_i).$$

This seemingly minor change has a major beneficial effect—the estimator becomes consistent and doubly robust.

**Theorem 1** (Informal). *Let* $\theta^\star$ *be defined as the minimizer* $\theta^\star = \arg\min_\theta \mathbb{E}_{(X,Y) \sim \mathbb{P}_X \times \mathbb{P}_{Y|X}}[\ell_\theta(X, Y)]$. *Under certain regularity conditions, we have*

$$\|\nabla_\theta \mathcal{L}^{\mathsf{DR}}_{\mathcal{D}_1, \mathcal{D}_2}(\theta^\star)\|_2 \lesssim \begin{cases} \sqrt{\frac{d}{m+n}}, & when \ Y \equiv \hat{f}(X), \\ \sqrt{\frac{d}{n}}, & otherwise. \end{cases}$$

*On the other hand, there exists instances such that* $\|\nabla_\theta \mathcal{L}^{\mathsf{SL}}_{\mathcal{D}_1, \mathcal{D}_2}(\theta^\star)\|_2 \geq C$ *always holds true no matter how large* $m, n$ *are.*

The result shows that the true parameter $\theta^\star$ is also a local minimum of the doubly robust loss, but not a local minimum of the original self-training loss. We flesh out this comparison for the special example of mean estimation in Section 2.1, and present empirical results on image and driving datasets in Section 3.

## 1.2 Related work

**Missing-data inference and causal inference.** The general problem of causal inference can be formulated as a missing-data inference problem as follows. For each unit in an experiment, at most one of the potential outcomes—the one corresponding to the treatment to which the unit is exposed—is observed, and the other potential outcomes are viewed as missing (Holland, 1986; Ding and Li, 2018). Two of the standard methods for solving this problem are data imputation Rubin (1979) and propensity score weighting Rosenbaum and Rubin (1983). A doubly robust causal inference estimator combines the virtues of these two methods. The estimator is referred to as "doubly robust" due to the following property: if the model for imputation is correctly specified then it is consistent no matter whether the propensity score model is correctly specified; on the other hand, if the model propensity score model is correctly specified, then it is consistent no matter whether the model for imputation is correctly specified (Scharfstein et al., 1999; Bang and Robins, 2005; Birhanu et al., 2011; Ding and Li, 2018).

We note in passing that double machine learning is another methodology that is inspired by the doubly robust paradigm in causal inference (Semenova et al., 2017; Chernozhukov et al., 2018a,b; Foster and Syrgkanis, 2019). The problem in double machine learning is related to the classic semiparametric problem of inference for a low-dimensional parameter in the presence of high-dimensional nuisance parameters, which is different goal than the predictive goal characterizing semi-supervised learning.

The recent work of prediction-powered inference (Angelopoulos et al., 2023) focuses on confidence estimation when there are both unlabeled data, labeled data, along with a teacher model. Their focus is the inferential problem of obtaining a confidence set, while ours is the doubly robust property of a point estimator. Since they focus on confidence estimation, an important, strong, yet biased baseline point-estimate algorithm that directly combines the ground-truth labels and pseudo-labels is not considered in their case. In our paper, we show with both theory and experiments that the proposed doubly-robust estimator achieves better performance than the naive combination of ground-truth labels and pseudo-labels.

**Self-training.** Self-training is a popular semi-supervised learning paradigm in which machine-generated pseudo-labels are used for training with unlabeled data (Lee, 2013; Berthelot et al., 2019b,a; Sohn et al., 2020a). To generate these pseudo-labels, a teacher model is pre-trained on a set of labeled data, and its predictions on the unlabeled data are extracted as pseudo-labels. Previous work seeks to address the noisy quality of pseudo-labels in various ways. MixMatch (Berthelot et al., 2019b) ensembles pseudo-labels across several augmented views of the input data. ReMixMatch (Berthelot et al., 2019a) extends this by weakly augmenting the teacher inputs and strongly augmenting the student inputs. FixMatch (Sohn et al., 2020a) uses confidence thresholding to select only high-quality pseudo-labels for student training.

Self-training has been applied in both 2D computer vision problems (Liu et al., 2021a; Jeong et al., 2019; Tang et al., 2021; Sohn et al., 2020b; Zhou et al., 2022) and 3D problems (Park et al., 2022; Wang et al., 2021; Li et al., 2023; Liu et al., 2023) object detection. STAC (Sohn et al., 2020b) enforces consistency between strongly augmented versions of confidence-filtered pseudo-labels. Unbiased teacher (Liu et al., 2021a) updates the teacher during training with an exponential moving average (EMA) of the student network weights. Dense Pseudo-Label (Zhou et al., 2022) replaces box pseudo-labels with the raw output features of the detector to allow the student to learn richer context. In the 3D domain, 3DIoUMatch (Wang et al., 2021) thresholds pseudo-labels using a model-predicted Intersection-over-Union (IoU). DetMatch (Park et al., 2022) performs detection in both the 2D and 3D domains and filters pseudo-labels based on 2D-3D correspondence. HSSDA (Liu et al., 2023) extends strong augmentation during training with a patch-based point cloud shuffling augmentation. Offboard3D (Qi et al., 2021) utilizes multiple frames of temporal context to improve pseudo-label quality.

There has been a limited amount of theoretical analysis of these methods, focusing on semi-supervised methods for mean estimation and linear regression (Zhang et al., 2019; Azriel et al., 2022). Our analysis bridges the gap between these analyses and the doubly robust estimators in the causal inference literature.

## 2 Doubly Robust Self-Training

We begin with the case where the marginal distributions for the covariates of the labeled and unlabeled datasets are the same. Assume that we are given a set of unlabeled samples, $\mathcal{D}_1 = \{X_1, X_2, \cdots, X_m\}$, drawn from a fixed distribution $\mathbb{P}_X$ supported on $\mathcal{X}$, a set of labeled samples $\mathcal{D}_2 = \{(X_{m+1}, Y_{m+1}), (X_{m+2}, Y_{m+2}), \cdots, (X_{m+n}, Y_{m+n})\}$, drawn from some joint distribution $\mathbb{P}_X \times \mathbb{P}_{Y|X}$ supported on $\mathcal{X} \times \mathcal{Y}$, and a pre-trained model, $\hat{f} : \mathcal{X} \mapsto \mathcal{Y}$. Let $\ell_\theta(\cdot, \cdot) : \mathcal{X} \times \mathcal{Y} \mapsto \mathbb{R}$ be a pre-specified loss function that characterizes the prediction error of the estimator with parameter $\theta$ on the given sample $(X, Y)$. Our target is to find some $\theta^\star \in \Theta$ that satisfies

$$\theta^\star \in \underset{\theta \in \Theta}{\arg\min}\, \mathbb{E}_{(X,Y)\sim\mathbb{P}_X \times \mathbb{P}_{Y|X}}[\ell_\theta(X, Y)].$$

For a given loss $\ell_\theta(x, y)$, consider a naive estimator that ignores the predictor $\hat{f}$ and only trains on the labeled samples:

$$\mathcal{L}^{\mathsf{TL}}_{\mathcal{D}_1, \mathcal{D}_2}(\theta) = \frac{1}{n} \sum_{i=m+1}^{m+n} \ell_\theta(X_i, Y_i).$$

Although naive, this is a safe choice since it is an empirical risk minimizer. As $n \to \infty$, the loss converges to the population loss. However, it ignores all the information provided in $\hat{f}$ and the unlabeled dataset, which makes it inefficient when the predictor $\hat{f}$ is informative.

On the other hand, traditional self-training aims at minimizing the combined loss for both labeled and unlabeled samples, where the pseudo-labels for unlabeled samples are generated using $\hat{f}$:[1]

$$\mathcal{L}^{\mathsf{SL}}_{\mathcal{D}_1, \mathcal{D}_2}(\theta) = \frac{1}{m+n}\left(\sum_{i=1}^{m} \ell_\theta(X_i, \hat{f}(X_i)) + \sum_{i=m+1}^{m+n} \ell_\theta(X_i, Y_i)\right)$$

$$= \frac{1}{m+n} \sum_{i=1}^{m+n} \ell_\theta(X_i, \hat{f}(X_i)) - \frac{1}{m+n} \sum_{i=m+1}^{m+n} \ell_\theta(X_i, \hat{f}(X_i)) + \frac{1}{m+n} \sum_{i=m+1}^{m+n} \ell_\theta(X_i, Y_i).$$

As is shown by the last equality, the self-training loss can be viewed as first using $\hat{f}$ to predict all the samples (including the labeled samples) and computing the average loss, then replacing that part of the loss corresponding to the labeled samples with the loss on the original labels. Although the loss uses the information arising from the unlabeled samples and $\hat{f}$, the performance can be poor when the predictor is not accurate.

We propose an alternative loss, which simply replaces the weight $1/(m+n)$ in the last two terms with $1/n$:

$$\mathcal{L}^{\mathsf{DR}}_{\mathcal{D}_1, \mathcal{D}_2}(\theta) = \frac{1}{m+n} \sum_{i=1}^{m+n} \ell_\theta(X_i, \hat{f}(X_i)) - \frac{1}{n} \sum_{i=m+1}^{m+n} \ell_\theta(X_i, \hat{f}(X_i)) + \frac{1}{n} \sum_{i=m+1}^{m+n} \ell_\theta(X_i, Y_i). \quad (1)$$

As we will show later, this is a doubly robust estimator. We provide an intuitive interpretation here:

- In the case when the given predictor is perfectly accurate, i.e., $\hat{f}(X) \equiv Y$ always holds (which also means that $Y|X = x$ is a deterministic function of $x$), the last two terms cancel, and the loss minimizes the average loss, $\frac{1}{m+n} \sum_{i=1}^{m+n} \ell_\theta(X_i, \hat{f}(X_i))$, on all of the provided data. The effective sample size is $m + n$, compared with effective sample size $n$ for training only on a labeled dataset using $\mathcal{L}^{\mathsf{TL}}$. In this case, the loss $\mathcal{L}^{\mathsf{DR}}$ is much better than $\mathcal{L}^{\mathsf{TL}}$, and comparable to $\mathcal{L}^{\mathsf{SL}}$.

  We may as well relax the assumption of $\hat{f}(X) = Y$ to $\mathbb{E}[\ell_\theta(X, \hat{f}(X))] = \mathbb{E}[\ell_\theta(X, Y)]$. As $n$ grows larger, the loss is approximately minimizing the average loss $\frac{1}{m+n} \sum_{i=1}^{m+n} \ell_\theta(X_i, \hat{f}(X_i))$.

---

[1]There are several variants of the traditional self-training loss. For example, Xie et al. (2020) introduce an extra weight $(m + n)/n$ on the labeled samples, and add noise to the student model; Sohn et al. (2020a) use confidence thresholding to filter unreliable pseudo-labels. However, both of these alternatives still suffer from the inconsistency issue. In this paper we focus on the simplest form $\mathcal{L}^{\mathsf{SL}}$.

- On the other hand, no matter how bad the given predictor is, the difference between the first two terms vanishes as either of $m, n$ goes to infinity since the labeled samples $X_{m+1}, \cdots, X_{m+n}$ arise from the same distribution as $X_1, \cdots, X_m$. Thus asymptotically the loss minimizes $\frac{1}{n} \sum_{i=m+1}^{m+n} \ell_\theta(X_i, Y_i)$, which discards the bad predictor $\hat{f}$ and focuses only on the labeled dataset. Thus, in this case the loss $\mathcal{L}^{\mathsf{DR}}$ is much better than $\mathcal{L}^{\mathsf{SL}}$, and comparable to $\mathcal{L}^{\mathsf{TL}}$.

This loss is appropriate only when the covariate distributions between labeled and unlabeled samples match. In the case where there is a distribution mismatch, we propose an alternative loss; see Section 2.3.

## 2.1 Motivating example: Mean estimation

As a concrete example, in the case of one-dimensional mean estimation we take $\ell_\theta(X, Y) = (\theta - Y)^2$. Our target is to find some $\theta^\star$ that satisfies
$$\theta^\star = \arg\min_\theta \mathbb{E}_{(X,Y) \sim \mathbb{P}_X \times \mathbb{P}_{Y|X}} [(\theta - Y)^2].$$
One can see that $\theta^\star = \mathbb{E}[Y]$. In this case, the loss for training only on labeled data becomes
$$\mathcal{L}^{\mathsf{TL}}_{\mathcal{D}_1, \mathcal{D}_2}(\theta) = \frac{1}{n} \sum_{i=m+1}^{m+n} (\theta - Y_i)^2.$$
Moreover, the optimal parameter is $\hat{\theta}_{\mathsf{TL}} = \frac{1}{n} \sum_{i=m+1}^{m+n} Y_i$, which is a simple empirical average over all observed $Y$'s.

For a given pre-existing predictor $\hat{f}$, the loss for self-training becomes
$$\mathcal{L}^{\mathsf{SL}}_{\mathcal{D}_1, \mathcal{D}_2}(\theta) = \frac{1}{m+n} \left( \sum_{i=1}^{m} (\theta - \hat{f}(X_i))^2 + \sum_{i=m+1}^{m+n} (\theta - Y_i)^2 \right).$$
It is straightforward to see that the minimizer of the loss is the unweighted average between the unlabeled predictors $\hat{f}(X_i)$'s and the labeled $Y_i$'s:
$$\theta^\star_{\mathsf{SL}} = \frac{1}{m+n} \left( \sum_{i=1}^{m} \hat{f}(X_i) + \sum_{i=m+1}^{m+n} Y_i \right).$$
In the case of $m \gg n$, the mean estimator is almost the same as the average of all the predicted values on the unlabeled dataset, which can be far from $\theta^\star$ when the predictor $\hat{f}$ is inaccurate.

On the other hand, for the proposed doubly robust estimator, we have
$$\mathcal{L}^{\mathsf{DR}}_{\mathcal{D}_1, \mathcal{D}_2}(\theta) = \frac{1}{m+n} \sum_{i=1}^{m+n} (\theta - \hat{f}(X_i))^2 - \frac{1}{n} \sum_{i=m+1}^{m+n} (\theta - \hat{f}(X_i))^2 + \frac{1}{n} \sum_{i=m+1}^{m+n} (\theta - Y_i)^2$$
$$= \frac{1}{m+n} \sum_{i=1}^{m+n} (\theta - \hat{f}(X_i))^2 + \frac{1}{n} \sum_{i=m+1}^{m+n} 2(\hat{f}(X_i) - Y_i)\theta + Y_i^2 - \hat{f}(X_i)^2.$$
Note that the loss is still convex, and we have
$$\theta^\star_{\mathsf{DR}} = \frac{1}{m+n} \sum_{i=1}^{m+n} \hat{f}(X_i) - \frac{1}{n} \sum_{i=m+1}^{m+n} (\hat{f}(X_i) - Y_i).$$
This recovers the estimator in prediction-powered inference (Angelopoulos et al., 2023). Assume that $\hat{f}$ is independent of the labeled data. We can calculate the mean-squared error of the three estimators as follows.

**Proposition 1.** *Let* $\mathsf{Var}[\hat{f}(X) - Y] = \mathbb{E}[(\hat{f}(X) - Y)^2 - \mathbb{E}[(\hat{f}(X) - Y)]^2]$. *We have*
$$\mathbb{E}[(\theta^\star - \hat{\theta}_{\mathsf{TL}})^2] = \frac{1}{n} \mathsf{Var}[Y],$$
$$\mathbb{E}[(\theta^\star - \hat{\theta}_{\mathsf{SL}})^2] \leq \frac{2m^2}{(m+n)^2} \mathbb{E}[(\hat{f}(X) - Y)]^2 + \frac{2m}{(m+n)^2} \mathsf{Var}[\hat{f}(X) - Y] + \frac{2n}{(m+n)^2} \mathsf{Var}[Y],$$
$$\mathbb{E}[(\theta^\star - \hat{\theta}_{\mathsf{DR}})^2] \leq 2 \min \left( \frac{1}{n} \mathsf{Var}[Y] + \frac{m+2n}{(m+n)n} \mathsf{Var}[\hat{f}(X)], \frac{m+2n}{(m+n)n} \mathsf{Var}[\hat{f}(X) - Y] + \frac{1}{m+n} \mathsf{Var}[Y] \right).$$

The proof is deferred to Appendix F. The proposition illustrates the double-robustness of $\hat{\theta}_{\text{DR}}$—no matter how poor the estimator $\hat{f}(X)$ is, the rate is always upper bounded by $\frac{4}{n}(\text{Var}[Y] + \text{Var}[\hat{f}(X)])$. On the other hand, when $\hat{f}(X)$ is an accurate estimator of $Y$ (i.e., $\text{Var}[\hat{f}(X) - Y]$ is small), the rate can be improved to $\frac{2}{m+n}\text{Var}[Y]$. In contrast, the self-training loss always has a non-vanishing term, $\frac{2m^2}{(m+n)^2}\mathbb{E}[(\hat{f}(X) - Y)]^2$, when $m \gg n$, unless the predictor $\hat{f}$ is accurate.

On the other hand, when $\hat{f}(x) = \hat{\beta}_{(-1)}^\top x + \hat{\beta}_1$ is a linear predictor trained on the labeled data with $\hat{\beta} = \arg\min_{\beta=[\beta_1,\beta_{(-1)}]} \frac{1}{n}\sum_{i=m+1}^{m+n}(\beta_{(-1)}^\top X_i + \beta_1 - Y_i)^2$, our estimator reduces to the semi-supervised mean estimator in Zhang et al. (2019). Let $\tilde{X} = [1, X]$. In this case, we also know that the self-training reduces to training only on labeled data, since $\hat{\theta}_{\text{TL}}$ is also the minimizer of the self-training loss. We have the following result that reveals the superiority of the doubly robust estimator compared to the other two options.

**Proposition 2** ((Zhang et al., 2019)). *We establish the asymptotic behavior of various estimators when $\hat{f}$ is a linear predictor trained on the labeled data:*

- *Training only on labeled data $\hat{\theta}_{\text{TL}}$ is equivalent to self-training $\hat{\theta}_{\text{SL}}$, which gives unbiased estimator but with larger variance:*

$$\sqrt{n}(\hat{\theta}_{\text{TL}} - \theta^\star) \to \mathcal{N}(0, \mathbb{E}[(Y - \beta^\top \tilde{X})^2] + \beta_{(-1)}^\top \Sigma \beta_{(-1)}).$$

- *Doubly Robust $\hat{\theta}_{\text{DR}}$ is unbiased with smaller variance:*

$$\sqrt{n}(\hat{\theta}_{\text{DR}} - \theta^\star) \to \mathcal{N}(0, \mathbb{E}[(Y - \beta^\top \tilde{X})^2] + \frac{n}{m+n}\beta_{(-1)}^\top \Sigma \beta_{(-1)}).$$

*Here $\beta = \arg\min_\beta \mathbb{E}[(Y - \beta^\top \tilde{X})^2]$ and $\Sigma = \mathbb{E}[(X - \mathbb{E}[X])(X - \mathbb{E}[X])^\top]$.*

## 2.2 Guarantee for general loss

In the general case, we show that the doubly robust loss function continues to exhibit desirable properties. In particular, as $n, m$ goes to infinity, the global minimum of the original loss is also a critical point of the new doubly robust loss, no matter how inaccurate the predictor $\hat{f}$.

Let $\theta^\star$ be the minimizer of $\mathbb{E}_{\mathbb{P}_{X,Y}}[\ell_\theta(X, Y)]$. Let $\hat{f}$ be a pre-existing model that does not depend on the datasets $\mathcal{D}_1, \mathcal{D}_2$. We also make the following regularity assumptions.

**Assumption 1.** *The loss $\ell_\theta(x, y)$ is differentiable at $\theta^\star$ for any $x, y$.*

**Assumption 2.** *The random variables $\nabla_\theta \ell_\theta(X, \hat{f}(X))$ and $\nabla_\theta \ell_\theta(X, Y)$ have bounded first and second moments.*

Given this assumption, we denote $\Sigma_\theta^{Y-\hat{f}} = \text{Cov}[\nabla_\theta \ell_\theta(X, \hat{f}(X)) - \nabla_\theta \ell_\theta(X, Y)]$ and let $\Sigma_\theta^{\hat{f}} = \text{Cov}[\nabla_\theta \ell_\theta(X, \hat{f}(X))]$, $\Sigma_\theta^Y = \text{Cov}[\nabla_\theta \ell_\theta(X, Y)]$.

**Theorem 2.** *Under Assumptions 1 and 2, we have that with probability at least $1 - \delta$,*

$$\|\nabla_\theta \mathcal{L}_{\mathcal{D}_1,\mathcal{D}_2}^{\text{DR}}(\theta^\star)\|_2 \leq C \min\left( \|\Sigma_{\theta^\star}^{\hat{f}}\|_2 \sqrt{\frac{d}{(m+n)\delta}} + \|\Sigma_{\theta^\star}^{Y-\hat{f}}\|_2 \sqrt{\frac{d}{n\delta}}, \right.$$

$$\left. \|\Sigma_{\theta^\star}^{\hat{f}}\|_2 \left( \sqrt{\frac{d}{(m+n)\delta}} + \sqrt{\frac{d}{n\delta}} \right) + \|\Sigma_{\theta^\star}^Y\|_2 \sqrt{\frac{d}{n\delta}} \right),$$

*where $C$ is a universal constant, and $\mathcal{L}_{\mathcal{D}_1,\mathcal{D}_2}^{\text{DR}}$ is defined in Equation (1).*

The proof is deferred to Appendix G. From the example of mean estimation we know that one can design instances such that $\|\nabla_\theta \mathcal{L}_{\mathcal{D}_1,\mathcal{D}_2}^{\text{SL}}(\theta^\star)\|_2 \geq C$ for some positive constant $C$.

When the loss $\nabla_\theta \mathcal{L}_{\mathcal{D}_1,\mathcal{D}_2}^{\text{DR}}$ is convex, the global minimum of $\nabla_\theta \mathcal{L}_{\mathcal{D}_1,\mathcal{D}_2}^{\text{DR}}$ converges to $\theta^\star$ as both $m, n$ go to infinity. When the loss $\nabla_\theta \mathcal{L}_{\mathcal{D}_1,\mathcal{D}_2}^{\text{DR}}$ is strongly convex, it also implies that $\|\hat{\theta} - \theta^\star\|_2$ converges to zero as both $m, n$ go to infinity, where $\hat{\theta}$ is the minimizer of $\nabla_\theta \mathcal{L}_{\mathcal{D}_1,\mathcal{D}_2}^{\text{DR}}$.

When $\hat{f}$ is a perfect predictor with $\hat{f}(X) \equiv Y$ (and $Y|X = x$ is deterministic), one has $\mathcal{L}^{\mathsf{DR}}_{\mathcal{D}_1,\mathcal{D}_2}(\theta^\star) = \frac{1}{m+n} \sum_{i=1}^{m+n} \ell_\theta(X_i, Y_i)$. The effective sample size is $m + n$ instead of $n$ in $\mathcal{L}^{\mathsf{SL}}_{\mathcal{D}_1,\mathcal{D}_2}(\theta)$.

When $\hat{f}$ is also trained from the labeled data, one may apply data splitting to achieve the same guarantee up to a constant factor. We provide further discussion in Appendix E.

## 2.3 The case of distribution mismatch

We also consider the case in which the marginal distributions of the covariates for the labeled and unlabeled datasets are different. Assume in particular that we are given a set of unlabeled samples, $\mathcal{D}_1 = \{X_1, X_2, \cdots, X_m\}$, drawn from a fixed distribution $\mathbb{P}_X$, a set of labeled samples, $\mathcal{D}_2 = \{(X_{m+1}, Y_{m+1}), (X_{m+2}, Y_{m+2}), \cdots, (X_{m+n}, Y_{m+n})\}$, drawn from some joint distribution $\mathbb{Q}_X \times \mathbb{P}_{Y|X}$, and a pre-trained model $\hat{f}$. In the case when the labeled samples do not follow the same distribution as the unlabeled samples, we need to introduce an importance weight $\pi(x)$. This yields the following doubly robust estimator:

$$\mathcal{L}^{\mathsf{DR2}}_{\mathcal{D}_1,\mathcal{D}_2}(\theta) = \frac{1}{m} \sum_{i=1}^{m} \ell_\theta(X_i, \hat{f}(X_i)) - \frac{1}{n} \sum_{i=m+1}^{m+n} \frac{1}{\pi(X_i)} \ell_\theta(X_i, \hat{f}(X_i)) + \frac{1}{n} \sum_{i=m+1}^{m+n} \frac{1}{\pi(X_i)} \ell_\theta(X_i, Y_i).$$

Note that we not only introduce the importance weight $\pi$, but we also change the first term from the average of all the $m + n$ samples to the average of $n$ samples.

**Proposition 3.** *We have* $\mathbb{E}[\mathcal{L}^{\mathsf{DR2}}_{\mathcal{D}_1,\mathcal{D}_2}(\theta)] = \mathbb{E}_{\mathbb{P}_{X,Y}}[\ell_\theta(X, Y)]$ *as long as one of the following two assumptions hold:*

- *For any $x$, $\pi(x) = \frac{\mathbb{P}_X(x)}{\mathbb{Q}_X(x)}$.*

- *For any $x$, $\ell_\theta(x, \hat{f}(x)) = \mathbb{E}_{Y \sim \mathbb{P}_{Y|X=x}}[\ell_\theta(x, Y)]$.*

The proof is deferred to Appendix H. The proposition implies that as long as either $\pi$ or $\hat{f}$ is accurate, the expectation of the loss is the same as that of the target loss. When the distributions for the unlabeled and labeled samples match each other, this reduces to the case in the previous sections. In this case, taking $\pi(x) = 1$ guarantees that the expectation of the doubly robust loss is always the same as that of the target loss.

# 3 Experiments

To employ the new doubly robust loss in practical applications, we need to specify an appropriate optimization procedure, in particular one that is based on (mini-batched) stochastic gradient descent so as to exploit modern scalable machine learning methods. In preliminary experiments we observed that directly minimizing the doubly robust loss in Equation (1) with stohastic gradient can lead to instability, and thus, we propose instead to minimize the curriculum-based loss in each epoch:

$$\mathcal{L}^{\mathsf{DR},t}_{\mathcal{D}_1,\mathcal{D}_2}(\theta) = \frac{1}{m+n} \sum_{i=1}^{m+n} \ell_\theta(X_i, \hat{f}(X_i)) - \alpha_t \cdot \left( \frac{1}{n} \sum_{i=m+1}^{m+n} \ell_\theta(X_i, \hat{f}(X_i)) - \frac{1}{n} \sum_{i=m+1}^{m+n} \ell_\theta(X_i, Y_i) \right).$$

As we show in the experiments below, this choice yields a stable algorithm. We set $\alpha_t = t/T$, where $T$ is the total number of epochs. For the object detection experiments, we introduce the labeled samples only in the final epoch, setting $\alpha_t = 0$ for all epochs before setting $\alpha_t = 1$ in the final epoch. Intuitively, we start from the training with samples only from the pseudo-labels, and gradually introduce the labeled samples in the doubly robust loss for fine-tuning.

We conduct experiments on both image classification task with ImageNet dataset (Russakovsky et al., 2015) and 3D object detection task with autonomous driving dataset nuScenes (Caesar et al., 2020). The code is available in `https://github.com/dingmyu/Doubly-Robust-Self-Training`.

## 3.1 Image classification

**Datasets and settings.** We evaluate our doubly robust self-training method on the ImageNet100 dataset, which contains a random subset of 100 classes from ImageNet-1k (Russakovsky et al., 2015),

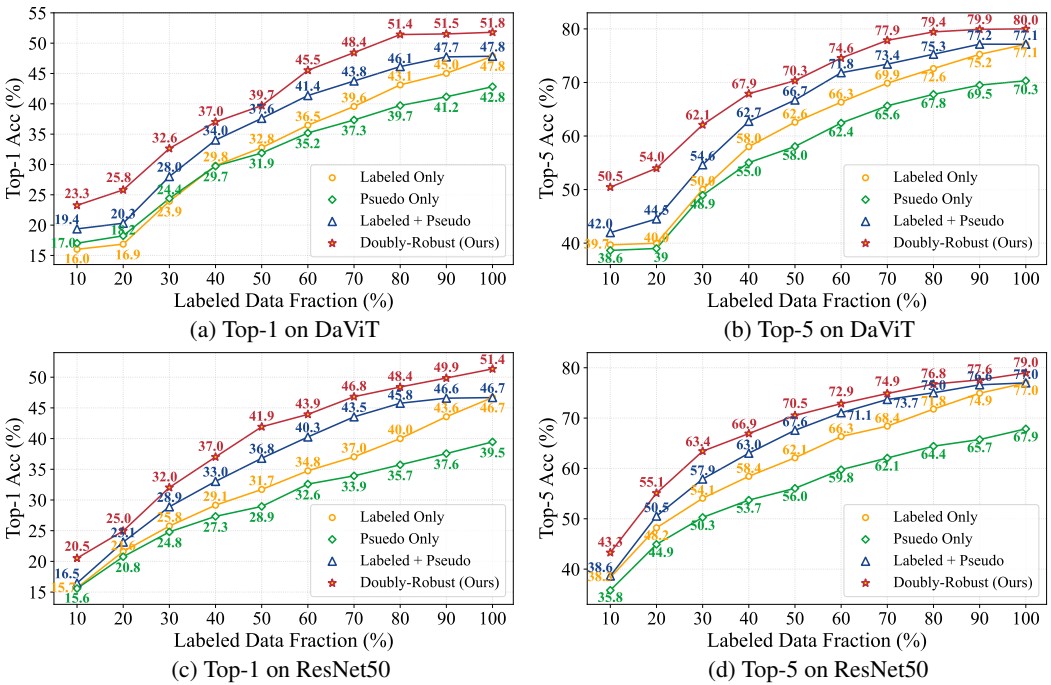

Figure 1: Comparisons on ImageNet100 using two different network architectures. Both Top-1 and Top-5 accuracies are reported. All models are trained for 20 epochs.

with 120K training images (approximately 1,200 samples per class) and 5,000 validation images (50 samples per class). To further test the effectiveness of our algorithm in a low-data scenario, we create a dataset that we refer to as mini-ImageNet100 by randomly sampling 100 images per class from ImageNet100. Two models were evaluated: (1) DaViT-T (Ding et al., 2022), a popular vision transformer architecture with state-of-the-art performance on ImageNet, and (2) ResNet50 (He et al., 2016), a classic convolutional network to verify the generality of our algorithm.

**Baselines.** To provide a comparative evaluation of doubly robust self-training, we establish three baselines: (1) 'Labeled Only' for training on labeled data only (partial training set) with a loss $\mathcal{L}^{\mathsf{TL}}$, (2) 'Pseudo Only' for training with pseudo labels generated for all training samples, and (3) 'Labeled + Pseudo' for a mixture of pseudo-labels and labeled data, with the loss $\mathcal{L}^{\mathsf{SL}}$. See the Appendix for further implementation details and ablations.

**Results on ImageNet100.** We first conduct experiments on ImageNet100 by training the model for 20 epochs using different fractions of labeled data from 1% to 100%. From the results shown in Fig. 1, we observe that: (1) Our model outperforms all baseline methods on both two networks by large margins. For example, we achieve 5.5% and 5.3% gains (Top-1 Acc) on DaViT over the 'Labeled + Pseudo' method for 20% and 80% labeled data, respectively. (2) The 'Labeled + Pseudo' method consistently beats the 'Labeled Only' baseline. (3) While 'Pseudo Only' works for smaller fractions of the labeled data (less than 30%) on DaViT, it is inferior to 'Labeled Only' on ResNet50.

**Results on mini-ImageNet100.** We also perform comparisons on mini-ImageNet100 to demonstrate the performance when the total data volume is limited. From the results in Table 1, we see our model generally outperforms all baselines. As the dataset size decreases and the number of training epochs increases, the gain of our algorithm becomes smaller. This is expected, as (1) the models are not adequately trained and thus have noise issues, and (2) there are an insufficient number of ground truth labels to compute the last term of our loss function. In extreme cases, there is only one labeled sample (1%) per class.

### 3.2   3D object detection

**Doubly robust object detection.** Given a visual representation of a scene, 3D object detection aims to generate a set of 3D bounding box predictions $\{b_i\}_{i\in[m+n]}$ and a set of corresponding class predictions $\{c_i\}_{i\in[m+n]}$. Thus, each single ground-truth annotation $Y_i \in Y$ is a set $Y_i = (b_i, c_i)$

Table 1: Comparisons on mini-ImageNet100, all models trained for 100 epochs.

| Labeled Data Percent | Labeled Only | | Pseudo Only | | Labeled + Pseudo | | Doubly robust Loss | |
|---|---|---|---|---|---|---|---|---|
| | top1 | top5 | top1 | top5 | top1 | top5 | top1 | top5 |
| 1 | 2.72 | 9.18 | **2.81** | 9.57 | 2.73 | 9.55 | 2.75 | **9.73** |
| 5 | 3.92 | 13.34 | 4.27 | 13.66 | 4.27 | 14.4 | **4.89** | **16.38** |
| 10 | 6.76 | 20.84 | 7.27 | 21.64 | 7.65 | 22.48 | **8.01** | **21.90** |
| 20 | 12.3 | 31.3 | 13.46 | 30.79 | **13.94** | **32.63** | 13.50 | 32.17 |
| 50 | 20.69 | 46.86 | 20.92 | 45.2 | 24.9 | 50.77 | **25.31** | **51.61** |
| 80 | 27.37 | 55.57 | 25.57 | 50.85 | 30.63 | 58.85 | **30.75** | **59.41** |
| 100 | 31.07 | 60.62 | 28.95 | 55.35 | **34.33** | 62.78 | 34.01 | **63.04** |

Table 2: Performance comparison on nuScenes *val* set.

| Labeled Data Fraction | Labeled Only | | Labeled + Pseudo | | Doubly robust Loss | |
|---|---|---|---|---|---|---|
| | mAP↑ | NDS↑ | mAP↑ | NDS↑ | mAP↑ | NDS↑ |
| 1/24 | 7.56 | 18.01 | 7.60 | 17.32 | **8.18** | **18.33** |
| 1/16 | 11.15 | 20.55 | 11.60 | 21.03 | **12.30** | **22.10** |
| 1/4 | 25.66 | 41.41 | **28.36** | **43.88** | 27.48 | 43.18 |

containing a box and a class. During training, the object detector is supervised with a sum of the box regression loss $\mathcal{L}_{loc}$ and the classification loss $\mathcal{L}_{cls}$, i.e. $\mathcal{L}_{obj} = \mathcal{L}_{loc} + \mathcal{L}_{cls}$.

In the self-training protocol for object detection, pseudo-labels for a given scene $X_i$ are selected from the labeler predictions $f(X_i)$ based on some user-defined criteria (typically the model's detection confidence). Unlike in standard classification or regression, $Y_i$ will contain a differing number of labels depending on the number of objects in the scene. Furthermore, the number of extracted pseudo-labels $f(X_i)$ will generally not be equal to the number of scene ground-truth labels $Y_i$ due to false positive/negative detections. Therefore it makes sense to express the doubly robust loss function in terms of the individual box labels as opposed to the scene-level labels. We define the doubly robust object detection loss as follows:

$$\mathcal{L}_{obj}^{\mathsf{DR}}(\theta) = \frac{1}{M + N_{ps}} \sum_{i=1}^{M+N_{ps}} \ell_\theta(X_i, f(X_i)) - \frac{1}{N_{ps}} \sum_{i=M+1}^{M+N_{ps}} \ell_\theta(X_i', f(X_i')) + \frac{1}{N} \sum_{i=M+1}^{M+N} \ell_\theta(X_i, Y_i),$$

where $M$ is the total number of pseudo-label boxes from the unlabeled split, $N$ is the total number of labeled boxes, $X_i'$ is the scene with pseudo-label boxes from the *labeled* split, and $N_{ps}$ is the total number of pseudo-label boxes from the *labeled* split. We note that the last two terms now contain summations over a differing number of boxes, a consequence of the discrepancy between the number of manually labeled boxes and pseudo-labeled boxes. Both components of the object detection loss (localization/classification) adopt this form of doubly robust loss.

**Dataset and setting.** To evaluate doubly robust self-training in the autonomous driving setting, we perform experiments on the large-scale 3D detection dataset nuScenes (Caesar et al., 2020). The nuScenes dataset is comprised of 1000 scenes (700 training, 150 validation and 150 test) with each frame containing sensor information from RGB camera, LiDAR, and radar scans. Box annotations are comprised of 10 classes, with the class instance distribution following a long-tailed distribution, allowing us to investigate our self-training approach for both common and rare classes. The main 3D detection metrics for nuScenes are mean Average Precision (mAP) and the nuScenes Detection Score (NDS), a dataset-specific metric consisting of a weighted average of mAP and five other true-positive metrics. For the sake of simplicity, we train object detection models using only LiDAR sensor information.

**Results.** After semi-supervised training, we evaluate our student model performance on the nuScenes *val* set. We compare three settings: training the student model with only the available labeled data

Table 3: Per-class mAP (%) comparison on nuScenes *val* set using 1/16 of total labels in training.

| | Car | Ped | Truck | Bus | Trailer | Barrier | Traffic Cone |
|---|---|---|---|---|---|---|---|
| Labeled Only | 48.6 | 30.6 | 8.5 | 6.2 | 4.0 | 6.8 | 4.4 |
| Labeled + Pseudo | 48.8 | 30.9 | 8.8 | 7.5 | 5.7 | 6.7 | 4.0 |
| Improvement | +0.2 | +0.3 | +0.3 | +1.3 | **+1.7** | -0.1 | -0.4 |
| Doubly robust Loss | 51.5 | 32.9 | 9.6 | 8.2 | 5.2 | 7.2 | 4.5 |
| Improvement | **+2.9** | **+2.3** | **+1.1** | **+2.0** | +1.2 | **+0.4** | **+0.1** |

(i.e., equivalent to teacher training), training the student model on the combination of labeled/teacher-labeled data using the naive self-training loss, and training the student model on the combination of labeled/teacher-labeled data using our proposed doubly robust loss. We report results for training with 1/24, 1/16, and 1/4 of the total labels in Table 2. We find that the doubly robust loss improves both mAP and NDS over using only labeled data and the naive baseline in the lower label regime, whereas performance is slightly degraded when more labels are available. Furthermore, we also show a per-class performance breakdown in Table 3. We find that the doubly robust loss consistently improves performance for both common (car, pedestrian) and rare classes. Notably, the doubly robust loss is even able to improve upon the teacher in classes for which pseudo-label training *decreases* performance when using the naive training (e.g., barriers and traffic cones).

## 4   Conclusions

We have proposed a novel doubly robust loss for self-training. Theoretically, we analyzed the double-robustness property of the proposed loss, demonstrating its statistical efficiency when the pseudo-labels are accurate. Empirically, we showed that large improvements can be obtained in both image classification and 3D object detection.

As a direction for future work, it would be interesting to understand how the doubly robust loss might be applied to other domains that have a missing-data aspect, including model distillation, transfer learning, and continual learning. It is also important to find practical and efficient algorithms when the labeled and unlabeled data do not match in distribution.

## Acknowledgements

Banghua Zhu and Jiantao Jiao are partially supported by NSF IIS-1901252, CIF-1909499 and CIF-2211209. Michael I. Jordan is partially supported by NSF IIS-1901252. Philip Jacobson is supported by the National Defense Science and Engineering Graduate (NDSEG) Fellowship. This work is also partially supported by Berkeley DeepDrive.

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

# A    Implementation Details for Image Classification

We evaluate our doubly robust self-training method on the ImageNet100 and mini-ImageNet100 datasets, which are subsets of ImageNet-1k from ImageNet Large Scale Visual Recognition Challenge 2012 (Russakovsky et al., 2015). Two models are evaluated: (1) DaViT-T (Ding et al., 2022), a state-of-the-art 12-layer vision transformer architecture with a patch size of 4, a window size of 7, and an embedding dim of 768, and (2) ResNet50 (He et al., 2016), a classic and powerful convolutional network with 50 layers and embedding dim 2048. We evaluate all the models on the same ImageNet100 validation set (50 samples per class). For the training, we use the same data augmentation and regularization strategies following the common practice in Liu et al. (2021b); Lin et al. (2017); Ding et al. (2022). We train all the models with a batch size of 1024 on 8 Tesla V100 GPUs (the batch size is reduced to 64 if the number of training data is less than 1000). We use AdamW (Loshchilov and Hutter, 2017) optimizer and a simple triangular learning rate schedule (Smith and Topin, 2019). The weight decay is set to 0.05 and the maximal gradient norm is clipped to 1.0. The stochastic depth drop rates are set to 0.1 for all models. During training, we crop images randomly to $224 \times 224$, while a center crop is used during evaluation on the validation set. We use a curriculum setting where the $\alpha_t$ grows linearly or quadratically from 0 to 1 throughout the training. To show the effectiveness of our method, we also compare model training with different curriculum learning settings and varying numbers of epochs.

# B    Additional Experiments in Image Classification

Table 4: Ablation study on different curriculum settings on ImageNet-100. All models are trained in 20 epochs.

| Methods | 30% GTs | | 50% GTs | | 70% GTs | | 90% GTs | |
|---|---|---|---|---|---|---|---|---|
| | top1 | top5 | top1 | top5 | top1 | top5 | top1 | top5 |
| Naive Labeled + Pseudo | 28.01 | 54.63 | 37.6 | 66.72 | 43.76 | 73.42 | 47.74 | 77.15 |
| doubly robust, $\alpha_t = 1$ | 28.43 | 56.65 | 38.06 | 67.18 | 43.22 | 73.18 | 48.52 | 77.21 |
| doubly robust, $\alpha_t = t/T$ (linear) | 30.87 | 60.98 | 40.18 | 71.06 | **46.60** | **75.80** | **50.44** | **78.88** |
| doubly robust, $\alpha_t = (t/T)^2$ (quadratic) | **31.15** | **61.29** | **40.86** | **71.14** | 45.50 | 75.11 | 49.64 | 77.77 |

**Ablation study on curriculum settings.** There are three options for the curriculum setting: 1) $\alpha_t = 1$ throughout the whole training, 2) grows linearly with training iterations $\alpha_t = t/T$, 3) grows quadratically with training iterations $\alpha_t = (t/T)^2$. From results in Table 4, we see: the first option achieves comparable performance with the 'Naive Labeled + Pseudo' baseline. Both the linear and quadratic strategies show significant performance improvements: the linear one works better when more labeled data is available, e.g., 70% and 90%, while the quadratic one prefers less labeled data, e.g. 30% and 50%.

Table 5: Ablation study on the number of epochs. All models are trained using 10% labeled data on ImageNet-100.

| Training epochs | Labeled Only | | Pseudo Only | | Labeled + Pseudo | | doubly robust Loss | |
|---|---|---|---|---|---|---|---|---|
| | top1 | top5 | top1 | top5 | top1 | top5 | top1 | top5 |
| 20 | 16.02 | 39.68 | 17.02 | 38.64 | 19.38 | 41.96 | **21.88** | **47.18** |
| 50 | 25.00 | 51.21 | 28.90 | 53.74 | 30.36 | 57.04 | **36.65** | **65.68** |
| 100 | 35.57 | 64.66 | 44.43 | **71.56** | 42.44 | 68.94 | **45.98** | 70.66 |

**Ablation Study on the Number of Epochs.** We conduct experiments on different training epochs. The results are shown in Table 5. Our model is consistently superior to the baselines. And we can observe the gain is larger when the number of training epochs is relatively small, e.g. 20 and 50.

**Fully trained results (300 epochs) on ImageNet-100.** In our original experiments, we mostly focus on a teacher model that is not super accurate, since our method reduces to the original pseudo-labeling when the teacher model is completely correct for all labels. In this experiment, we fully train the teacher model with 300 epochs on ImageNet-100, leading to the accuracy of the teacher model as high as 88.0%. From Figure 2, we show that even in this case, our method outperforms the original pseudo-labeling baseline.

Figure 2: Results on ImageNet-100 using fully trained (300 epochs) DaViT-T with different data fractions.

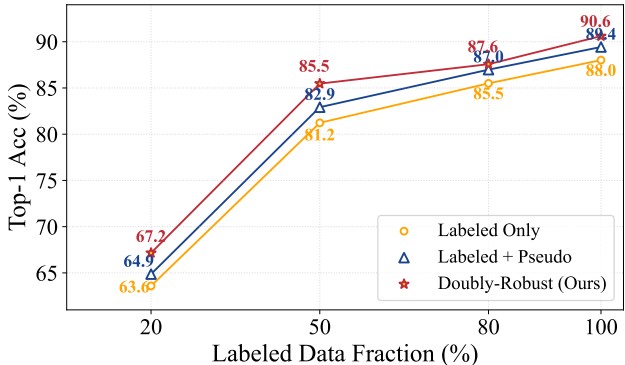

**Comparisons with previous SOTAs on CIFAR-10 and CiFAR-100.** We compare with another 11 baselines in terms of error rate on CIFAR-10-4K and CIFAR-100-10K under the same settings (i.e., Wide ResNet-28-2 for CIFAR-10 and WRN-28-8 for CIFAR-100). We show that our method is only 0.04 inferior to the best method Meta Pseudo Labels for CIFAR-10-4K, and achieves the best performance for CIFAR-100-10K.

Table 6: Comparisons with previous SOTAs on CiFAR-10 and CIFAR-100.

| Method | CIFAR-10-4K (error rate, %) | CIFAR-100-10K (error rate, %) |
|---|---|---|
| Pseudo-Labeling | 16.09 | 36.21 |
| LGA + VAT | 12.06 | – |
| Mean Teacher | 9.19 | 35.83 |
| ICT | 7.66 | – |
| SWA | 5.00 | 28.80 |
| MixMatch | 4.95 | 25.88 |
| ReMixMatch | 4.72 | 23.03 |
| EnAET | 5.35 | – |
| UDA | 4.32 | 24.50 |
| FixMatch | 4.31 | 23.18 |
| Meta Pesudo Labels | **3.89** | – |
| **Ours** | 3.93 | **22.30** |

# C   Implementation Details of 3D Object Detection

Our experiments follow the standard approach for semi-supervised detection: we first initialize two detectors, the teacher (i.e., labeler) and the student. First, a random split of varying sizes is selected from the nuScenes training set. We pre-train the teacher network using the ground-truth annotations in this split. Following this, we freeze the weights in the teacher model and then use it to generate pseudo-labels on the entire training set. The student network is then trained on a combination of the pseudo-labels and ground-truth labels originating from the original split. In all of our semi-supervised experiments, we use CenterPoint with a PointPillars backbone as our 3D detection model (Yin et al., 2021; Lang et al., 2019). The teacher pre-training and student training are both conducted for 10 epochs on 3 NVIDIA RTX A6000 GPUs. We follow the standard nuScenes training setting outlined in Zhu et al. (2019), with the exception of disabling ground-truth paste augmentation during training to prevent data leakage from the labeled split. To select the pseudo-labels to be used in training the student, we simply filter the teacher predictions by detection confidence, using all detections above a chosen threshold. We use a threshold of 0.3 for all classes, as in Park et al. (2022). In order to conduct training in a batch-wise manner, we compute the loss over only the samples contained within

the batch. We construct each batch to have a consistent ratio of labeled/unlabeled samples to ensure the loss is well-defined for the batch.

# D   Additional Experiments in 3D Object Detection

**Comparison with Semi-Supervised Baseline** We compare our approach to another semi-supervised baseline on the 3D object detection task, Pseudo-labeling and confirmation bias Arazo et al. (2020). We shows that in multiple settings, our approach surpasses the baseline performance.

**Ablation on Pseudo-label Confidence Threshold** To demonstrate that appropriate quality pseudo-labels are used to train the student detector, we performance ablation experiments varying the detection threshold used to extract pseudo-labels from the teacher model predictions. We show that training with a threshold of $\tau = 0.3$ outperforms training with a more stringent threshold, and is the appropriate experimental setting for our main experiments.

Table 7: Performance comparison with pseudo-labeling baseline on nuScenes *val* set.

| Labeled Fraction | Labeled Only mAP↑ | NDS↑ | Labeled + Pseudo mAP↑ | NDS↑ | Doubly robust Loss mAP↑ | NDS↑ | Pseudo-Labeling + Confirmation Bias mAP↑ | NDS↑ |
|---|---|---|---|---|---|---|---|---|
| 1/24 | 7.56 | 18.01 | 7.60 | 17.32 | **8.18** | **18.33** | 7.80 | 16.86 |
| 1/16 | 11.15 | 20.55 | 11.60 | 21.03 | **12.30** | 22.10 | 12.15 | **22.89** |

Table 8: Doubly Robust Loss performance comparison with differing detection thresholds for pseudo-labels.

| Labeled Data Fraction | $\tau = 0.3$ Labeled+Pseudo mAP↑ | NDS↑ | Doubly Robust Loss mAP↑ | NDS↑ | $\tau = 0.5$ Labeled+Pseudo mAP↑ | NDS↑ | Doubly Robust Loss mAP↑ | NDS↑ |
|---|---|---|---|---|---|---|---|---|
| 1/24 | 7.56 | 18.01 | **8.18** | **18.33** | 7.15 | 15.82 | 4.37 | 13.17 |
| 1/16 | 11.15 | 20.55 | **12.30** | **22.10** | 11.05 | 21.22 | 8.09 | 19.70 |

# E   Considerations when $\hat{f}$ is trained from labeled data

In Theorem 2, we analyzed the double robustness of the proposed loss function when the predictor $\hat{f}$ is pre-existing and not trained from the labeled dataset. In practice, one may only have access to the labeled and unlabeled datasets without a pre-existing teacher model. In this case, one may choose to split the labeled samples $\mathcal{D}_2$ into two parts. The last $n/2$ samples are used to train $\hat{f}$, and the first $n/2$ samples are used in the doubly robust loss:

$$\mathcal{L}^{\mathsf{DR2}}_{\mathcal{D}_1,\mathcal{D}_2}(\theta) = \frac{1}{m}\sum_{i=1}^{m}\ell_\theta(X_i,\hat{f}(X_i)) - \frac{2}{n}\sum_{i=m+1}^{m+n/2}\frac{1}{\pi(X_i)}\ell_\theta(X_i,\hat{f}(X_i)) + \frac{2}{n}\sum_{i=m+1}^{m+n/2}\frac{1}{\pi(X_i)}\ell_\theta(X_i,Y_i).$$

Since $\hat{f}$ is independent of all samples used in the above loss, the result in Theorem 2 continues to hold. Asymptotically, such a doubly robust estimator is never worse than the estimator trained only on the labeled data.

# F   Proof of Proposition 1

For the labeled-only estimator $\hat{\theta}_{\mathsf{TL}}$, we have

$$\mathbb{E}[(\theta^\star - \hat{\theta}_{\mathsf{TL}})^2] = \mathbb{E}\left[\left(\mathbb{E}[Y] - \frac{1}{n}\sum_{i=m+1}^{m+n}Y\right)^2\right] = \frac{1}{n}\mathsf{Var}[Y].$$

For the self-training loss, we have

$$\mathbb{E}[(\theta^\star - \hat{\theta}_{\mathsf{SL}})^2] = \mathbb{E}\left[\left(\mathbb{E}[Y] - \frac{1}{m+n}\left(\sum_{i=1}^{m}\hat{f}(X_i) + \sum_{i=m+1}^{m+n}Y_i\right)\right)^2\right]$$

$$\leq 2\left(\mathbb{E}\left[\left(\frac{m}{m+n}\left(\mathbb{E}[Y] - \frac{1}{m}\sum_{i=1}^{m}\hat{f}(X_i)\right)\right)^2\right] + \mathbb{E}\left[\left(\frac{n}{m+n}\left(\mathbb{E}[Y] - \frac{1}{n}\sum_{i=m+1}^{m+n}Y_i\right)\right)^2\right]\right)$$

$$\leq \frac{2m^2}{(m+n)^2}\mathbb{E}[(\hat{f}(X) - Y)]^2 + \frac{2m}{(m+n)^2}\mathsf{Var}[\hat{f}(X) - Y] + \frac{2n}{(m+n)^2}\mathsf{Var}[Y].$$

For the doubly robust loss, on one hand, we have

$$\mathbb{E}[(\theta^\star - \hat{\theta}_{\mathsf{DR}})^2] = \mathbb{E}\left[\left(\mathbb{E}[Y] - \frac{1}{m+n}\sum_{i=1}^{m+n}\hat{f}(X_i) + \frac{1}{n}\sum_{i=m+1}^{m+n}(\hat{f}(X_i) - Y_i)\right)^2\right]$$

$$\leq 2\mathbb{E}\left[\left(\mathbb{E}[Y] - \frac{1}{n}\sum_{i=m+1}^{m+n}Y_i\right)^2\right] + 2\mathbb{E}\left[\left(\mathbb{E}[\hat{f}(X)] - \frac{1}{n}\sum_{i=m+1}^{m+n}\hat{f}(X_i)\right)^2\right]$$

$$+ 2\mathbb{E}\left[\left(\mathbb{E}[\hat{f}(X)] - \frac{1}{m+n}\sum_{i=1}^{m+n}\hat{f}(X_i)\right)^2\right]$$

$$= \frac{2}{n}\mathsf{Var}[Y] + \left(\frac{2}{m+n} + \frac{2}{n}\right)\mathsf{Var}[\hat{f}(X)].$$

On the other hand, we have

$$\mathbb{E}[(\theta^\star - \hat{\theta}_{\mathsf{DR}})^2] = \mathbb{E}\left[\left(\mathbb{E}[Y] - \frac{1}{m+n}\sum_{i=1}^{m+n}\hat{f}(X_i) + \frac{1}{n}\sum_{i=m+1}^{m+n}(\hat{f}(X_i) - Y_i)\right)^2\right]$$

$$\leq 2\mathbb{E}\left[\left(\mathbb{E}[Y] - \frac{1}{m+n}\sum_{i=1}^{m+n}Y_i\right)^2\right] + 2\mathbb{E}\left[\left(\mathbb{E}[\hat{f}(X) - Y] - \frac{1}{n}\sum_{i=m+1}^{m+n}(\hat{f}(X_i) - Y_i)\right)^2\right]$$

$$+ 2\mathbb{E}\left[\left(\mathbb{E}[\hat{f}(X) - Y] - \frac{1}{m+n}\sum_{i=1}^{m+n}(\hat{f}(X_i) - Y_i)\right)^2\right]$$

$$= \left(\frac{2}{m+n} + \frac{2}{n}\right)\mathsf{Var}[\hat{f}(X) - Y] + \frac{2}{m+n}\mathsf{Var}[Y].$$

The proof is done by taking the minimum of the two upper bounds.

## G  Proof of Theorem 2

*Proof.* We know that

$$\|\nabla_\theta \mathcal{L}_{\mathcal{D}_1,\mathcal{D}_2}^{\mathsf{DR}}(\theta^\star) - \mathbb{E}[\nabla_\theta \mathcal{L}_{\mathcal{D}_1,\mathcal{D}_2}^{\mathsf{DR}}(\theta^\star)]\|_2$$

$$= \left\|\frac{1}{m+n}\sum_{i=1}^{m+n}(\nabla_\theta \ell_{\theta^\star}(X_i, \hat{f}(X_i)) - \mathbb{E}[\nabla_\theta \ell_{\theta^\star}(X, \hat{f}(X))]) + \frac{1}{n}\sum_{i=m+1}^{m+n}\left(\nabla_\theta \ell_{\theta^\star}(X_i, Y_i) - \nabla_\theta \ell_{\theta^\star}(X_i, \hat{f}(X_i))\right.\right.$$

$$\left.\left. - \mathbb{E}[\nabla_\theta \ell_{\theta^\star}(X, Y) - \nabla_\theta \ell_{\theta^\star}(X, \hat{f}(X))]\right)\right\|_2$$

$$\leq \left\|\frac{1}{m+n}\sum_{i=1}^{m+n}(\nabla_\theta \ell_{\theta^\star}(X_i, \hat{f}(X_i)) - \mathbb{E}[\nabla_\theta \ell_{\theta^\star}(X, \hat{f}(X))])\right\|_2 + \left\|\frac{1}{n}\sum_{i=m+1}^{m+n}\left(\nabla_\theta \ell_{\theta^\star}(X_i, Y_i) - \nabla_\theta \ell_{\theta^\star}(X_i, \hat{f}(X_i))\right.\right.$$

$$\left.\left. - \mathbb{E}[\nabla_\theta \ell_{\theta^\star}(X, Y) - \nabla_\theta \ell_{\theta^\star}(X, \hat{f}(X))]\right)\right\|_2.$$

From the multi-dimensional Chebyshev inequality (Bibby et al., 1979; Marshall and Olkin, 1960), we have that with probability at least $1 - \delta/2$, for some universal constant $C$,

$$\left\| \frac{1}{m+n} \sum_{i=1}^{m+n} (\nabla_\theta \ell_{\theta^\star}(X_i, \hat{f}(X_i)) - \mathbb{E}[\nabla_\theta \ell_{\theta^\star}(X, \hat{f}(X))]) \right\|_2 \leq C \|\Sigma_{\theta^\star}^{\hat{f}}\|_2 \sqrt{\frac{d}{(m+n)\delta}}.$$

Similarly, we also have that with probability at least $1 - \delta/2$,

$$\left\| \frac{1}{n} \sum_{i=m+1}^{m+n} \left( \nabla_\theta \ell_{\theta^\star}(X_i, Y_i) - \nabla_\theta \ell_{\theta^\star}(X_i, \hat{f}(X_i)) - \mathbb{E}[\nabla_\theta \ell_{\theta^\star}(X, Y) - \nabla_\theta \ell_{\theta^\star}(X, \hat{f}(X))] \right) \right\|_2 \leq C \|\Sigma_{\theta^\star}^{Y-\hat{f}}\|_2 \sqrt{\frac{d}{n\delta}}.$$

Furthermore, note that

$$\mathbb{E}[\nabla_\theta \mathcal{L}_{\mathcal{D}_1, \mathcal{D}_2}^{\mathsf{DR}}(\theta^\star)] = \mathbb{E}[\nabla_\theta \ell_{\theta^\star}(X, Y)] = \nabla_\theta \mathbb{E}[\ell_{\theta^\star}(X, Y)] = 0.$$

Here we use Assumption 1 and Assumption 2 to ensure that the expectation and differentiation are interchangeable. Thus we have that with probability at least $1 - \delta$,

$$\|\nabla_\theta \mathcal{L}_{\mathcal{D}_1, \mathcal{D}_2}^{\mathsf{DR}}(\theta^\star)\|_2 \leq C \left( \|\Sigma_{\theta^\star}^{\hat{f}}\|_2 \sqrt{\frac{d}{(m+n)\delta}} + \|\Sigma_{\theta^\star}^{Y-\hat{f}}\|_2 \sqrt{\frac{d}{n\delta}} \right).$$

On the other hand, we can also write the difference as

$$\|\nabla_\theta \mathcal{L}_{\mathcal{D}_1, \mathcal{D}_2}^{\mathsf{DR}}(\theta^\star) - \mathbb{E}[\nabla_\theta \mathcal{L}_{\mathcal{D}_1, \mathcal{D}_2}^{\mathsf{DR}}(\theta^\star)]\|_2$$

$$= \left\| \frac{1}{m+n} \sum_{i=1}^{m+n} (\nabla_\theta \ell_{\theta^\star}(X_i, \hat{f}(X_i)) - \mathbb{E}[\nabla_\theta \ell_{\theta^\star}(X, \hat{f}(X))]) + \frac{1}{n} \sum_{i=m+1}^{m+n} \left( \nabla_\theta \ell_{\theta^\star}(X_i, Y_i) - \mathbb{E}[\nabla_\theta \ell_{\theta^\star}(X, Y)] \right) \right.$$

$$\left. - \frac{1}{n} \sum_{i=m+1}^{m+n} \left( \nabla_\theta \ell_{\theta^\star}(X_i, Y_i) - \mathbb{E}[\nabla_\theta \ell_{\theta^\star}(X, \hat{f}(X))] \right) \right\|_2$$

$$\leq \left\| \frac{1}{m+n} \sum_{i=1}^{m+n} (\nabla_\theta \ell_{\theta^\star}(X_i, \hat{f}(X_i)) - \mathbb{E}[\nabla_\theta \ell_{\theta^\star}(X, \hat{f}(X))]) \right\|_2 + \left\| \frac{1}{n} \sum_{i=m+1}^{m+n} \left( \nabla_\theta \ell_{\theta^\star}(X_i, Y_i) - \mathbb{E}[\nabla_\theta \ell_{\theta^\star}(X, Y)] \right) \right\|_2$$

$$+ \left\| \frac{1}{n} \sum_{i=m+1}^{m+n} \left( \nabla_\theta \ell_{\theta^\star}(X_i, Y_i) - \mathbb{E}[\nabla_\theta \ell_{\theta^\star}(X, \hat{f}(X))] \right) \right\|_2$$

$$\leq C \left( \|\Sigma_{\theta^\star}^{\hat{f}}\|_2 \left( \sqrt{\frac{d}{(m+n)\delta}} + \sqrt{\frac{d}{n\delta}} \right) + \|\Sigma_{\theta^\star}^{Y}\|_2 \sqrt{\frac{d}{n\delta}} \right).$$

Here the last inequality uses the multi-dimensional Chebyshev inequality and it holds with probability at least $1 - \delta$. This finishes the proof. $\qquad \square$

## H  Proof of Proposition 3

*Proof.* We have

$$\mathbb{E}[\mathcal{L}_{\mathcal{D}_1, \mathcal{D}_2}^{\mathsf{DR2}}(\theta)] = \frac{1}{m} \sum_{i=1}^{m} \mathbb{E}_{X_i \sim \mathbb{P}_X}[\ell_\theta(X_i, \hat{f}(X_i))] - \frac{1}{n} \sum_{i=m+1}^{m+n} \mathbb{E}_{X_i \sim \mathbb{Q}_X} \left[ \frac{1}{\pi(X_i)} \ell_\theta(X_i, \hat{f}(X_i)) \right]$$

$$+ \frac{1}{n} \sum_{i=m+1}^{m+n} \mathbb{E}_{X_i \sim \mathbb{Q}_X, Y_i \sim \mathbb{P}_{Y|X_i}} \left[ \frac{1}{\pi(X_i)} \ell_\theta(X_i, Y_i) \right]$$

$$= \mathbb{E}_{X \sim \mathbb{P}_X}[\ell_\theta(X, \hat{f}(X))] - \mathbb{E}_{X \sim \mathbb{Q}_X} \left[ \frac{1}{\pi(X)} \ell_\theta(X, \hat{f}(X)) \right]$$

$$+ \mathbb{E}_{X \sim \mathbb{Q}_X, Y \sim \mathbb{P}_{Y|X}} \left[ \frac{1}{\pi(X)} \ell_\theta(X, Y) \right].$$

In the first case when $\pi(x) \equiv \frac{\mathbb{P}_X(x)}{\mathbb{Q}_X(x)}$, we have

$$
\begin{aligned}
\mathbb{E}[\mathcal{L}^{\mathsf{DR2}}_{\mathcal{D}_1,\mathcal{D}_2}(\theta)] &= \mathbb{E}_{X \sim \mathbb{P}_X}[\ell_\theta(X, \hat{f}(X))] - \mathbb{E}_{X \sim \mathbb{Q}_X}\left[\frac{\mathbb{P}_X(X)}{\mathbb{Q}_X(X)}\ell_\theta(X, \hat{f}(X))\right] \\
&\quad + \mathbb{E}_{X \sim \mathbb{Q}_X, Y \sim \mathbb{P}_{Y|X}}\left[\frac{\mathbb{P}_X(X)}{\mathbb{Q}_X(X)}\ell_\theta(X, Y)\right] \\
&= \mathbb{E}_{X \sim \mathbb{P}_X}[\ell_\theta(X, \hat{f}(X))] - \mathbb{E}_{X \sim \mathbb{P}_X}\left[\ell_\theta(X, \hat{f}(X))\right] \\
&\quad + \mathbb{E}_{X \sim \mathbb{P}_X, Y \sim \mathbb{P}_{Y|X}}[\ell_\theta(X, Y)] \\
&= \mathbb{E}_{X, Y \sim \mathbb{P}_{X,Y}}[\ell_\theta(X, Y)].
\end{aligned}
$$

In the second case when $\ell_\theta(x, \hat{f}(x)) = \mathbb{E}_{Y \sim \mathbb{P}_{Y|X=x}}[\ell_\theta(x, Y)]$, we have

$$
\begin{aligned}
\mathbb{E}[\mathcal{L}^{\mathsf{DR2}}_{\mathcal{D}_1,\mathcal{D}_2}(\theta)] &= \mathbb{E}_{X \sim \mathbb{P}_X}[\ell_\theta(X, \hat{f}(X))] - \mathbb{E}_{X \sim \mathbb{Q}_X}\left[\frac{1}{\pi(X)}\ell_\theta(X, \hat{f}(X))\right] \\
&\quad + \mathbb{E}_{X \sim \mathbb{Q}_X}\mathbb{E}_{Y \sim \mathbb{P}_{Y|X}}\left[\frac{1}{\pi(X)}\ell_\theta(X, Y) \mid X\right] \\
&= \mathbb{E}_{X \sim \mathbb{P}_X}[\ell_\theta(X, \hat{f}(X))] - \mathbb{E}_{X \sim \mathbb{Q}_X}\left[\frac{1}{\pi(X)}\ell_\theta(X, \hat{f}(X))\right] \\
&\quad + \mathbb{E}_{X \sim \mathbb{Q}_X}\left[\frac{1}{\pi(X)}\ell_\theta(X, \hat{f}(X))\right] \\
&= \mathbb{E}_{X \sim \mathbb{P}_X}[\ell_\theta(X, \hat{f}(X))] \\
&= \mathbb{E}_{X, Y \sim \mathbb{P}_{X,Y}}[\ell_\theta(X, Y)].
\end{aligned}
$$

This finishes the proof. $\qquad\square$

