# OpenReview forum: "Doubly-Robust Self-Training"
_NeurIPS.cc/2023/Conference — NeurIPS 2023 poster_

### Official Review · Reviewer_8nT5 · 2023-06-26

**Soundness:** 2 fair
**Presentation:** 3 good
**Contribution:** 2 fair
**Rating:** 4
**Confidence:** 4

**Summary:**

This paper presents a very simple approach to semi-supervised learning that utilizes both labeled and unlabeled datasets. When there is a large amount of unlabeled data available, following the same distribution as the labeled dataset, the most effective method to leverage this unlabeled data for training is self-training, where pseudo labels are generated and used for training. The main limitation of self-training is that the performance can degrade when the pseudo-labels produced by the predictor for the unlabeled set are not accurate.
In this paper, the authors propose a very simple method to overcome this limitation by replacing the conventional loss for the labeled dataset and the unlabeled dataset in self-training. This modification allows the model to be trained effectively in both cases, whether the pseudo-labels are correct or not, leading to improved performance in all scenarios. The effectiveness of this approach is experimentally demonstrated on image classification benchmarks and 3D object detection benchmarks.

**Strengths:**

Overall, the method is very simple, intuitive, and quite novel. The paper is well-written and it provides a well-derived explanation using equations for both cases where the pseudo labels are accurate and when they are not. Additionally, the method is thoroughly analyzed from a theoretical perspective.

**Weaknesses:**

Although the theoretical derivation demonstrates the soundness of the method, there is doubt regarding its effectiveness in experiments.

1. Especially, in the experiments (sec. 3.1), the authors use curriculum-based loss in each epoch. With $\alpha_t  < 1 $, the proposed method will behave like self-training (exactly when $\alpha_t = \frac{n}{m+n}$) and the behavior/effectiveness of the proposed method could not be well represented in this case.

2. Image classification experiments were conducted using ImageNet-100. However, when the labeled set ratio is 100%, the top-1 accuracy of DaVIT and ResNet50 is remarkably lower at 47.8% and 46.7%, respectively, compared to the reported top-1 accuracy in other papers. Following the paper of DaVit and ResNet, the top-1 accuracy on the more complex task of ImageNet1k, compared to ImageNet-100, using DaVit-tiny and ResNet is 82.8% and 79.26%, respectively. Following [1], the top-1 accuracy of various resnet50-based methods for ImageNet-100 consistently surpasses 70%. It appears that the baseline has not been sufficiently well-trained.

3. There is a lack of comparison with other semi-supervised methods. While this method compares with the basic self-training loss, it is necessary to compare it with various methods that utilize pseudo-labels for self-training [2], [3], [4]. Most semi-supervised image classification methods have performed experiments on benchmark datasets such as ImageNet1k and CIFAR100. These methods achieve a top-1 accuracy of 72% or higher when using a labeled dataset of 10% on ImageNet1k. Therefore, it is necessary to compare the effectiveness of this method on these benchmark datasets.

4. Similarly, in semi-supervised 3D object detection, it is necessary to compare the results with existing baseline methods [5], [6].

5. Although this paper proposes a method for the distribution mismatch case in Section 2.4, estimating the probability values of each sample, p(x) and q(x), for each distribution is not easy. Practical methods for this issue have not been provided, and experiments on this aspect are also lacking. Practical algorithms and experiments need to be provided in order to address this concern.

[1] Zelin Zang, et al. DLME: Deep Local-flatness Manifold Embedding. ECCV 2022

[2] Qizhe Xie, Zihang Dai, Eduard Hovy, Minh-Thang Luong, and Quoc V. Le. Unsupervised data augmentation for consistency training. In Advances in Neural Information Processing Systems, 2020

[3] Kihyuk Sohn, David Berthelot, Zizhao Li, Chun-Liang Zhang, Nicholas Carlini, Ekin D. Cubuk, Alex Kurakin, Han Zhang, and Colin Raffel. Fixmatch: Simplifying semi-supervised learning with consistency and confidence. In IEEE Conference on Computer Vision and Pattern Recognition, 2020

[4]  Hieu Pham, Qizhe Xie, Zihang Dai, and Quoc V Le. Meta pseudo labels. In IEEE Conference on Computer Vision and Pattern Recognition, 2021

[5] He Wang, Yezhen Cong, Or Litany, Yue Gao and Leonidas J. Guibas. 3DIoUMatch: Leveraging IoU Prediction for Semi-Supervised 3D
Object Detection. In IEEE Conference on Computer Vision and Pattern Recognition, 2021

[6] Na Zhao, Tat-Seng Chua and Gim Hee Lee. SESS: Self-Ensembling Semi-Supervised 3D Object Detection. In IEEE Conference on Computer Vision and Pattern Recognition, 2020

**Questions:**

see above.

**Limitations:**

Future work is discussed.

---

> ### Author Rebuttal · Authors · 2023-08-06
>
> Thank you for your valuable comments and suggestions. We have corrected all the typos suggested. Please find our responses to each comment below.
>
> ## Comment 1
>
> **Reviewer:**
>
>
> > In the experiments (sec. 3.1), the authors use curriculum-based loss in each epoch. With $\alpha_t<1$, the proposed method will behave like self-training (exactly when $\alpha_t = \frac{n}{m+n}$) and the behavior/effectiveness of the proposed method could not be well represented in this case.
>
> **Response:**
>
> Thank you for your comments! Our curriculum-based loss only matches the original self-training loss when exactly $\alpha_t=0$ (which reduces to $\frac{1}{m+n}\sum_{i=1}^{m+n} \ell_\theta(X_i, \hat f(X_i))$). For any $\alpha_t>0$, it is an interpolation between the original self-training loss and our proposed loss function. Our experimental results also demonstrate a significant improvement over the original self-training loss with $\alpha_t$ always equal to $0$.
>
> We would like to also clarify that even when $\alpha_t = \frac{n}{m+n}$, this loss is **very different from self-training loss**. In this case, our loss becomes $\frac{1}{m+n}\sum_{i=1}^{m+n} \ell_\theta(X_i, \hat f(X_i)) - \frac{1}{m+n}\sum_{i=m+1}^{m+n} \ell_\theta(X_i, \hat f(X_i)) + \frac{1}{m+n}\sum_{i=m+1}^{m+n} \ell_\theta(X_i, Y_i)$. Note that none of the terms cancel since the second and third term is averaging over the labeled samples and reweighted, while the first term is average over both label and unlabeled samples. In contrast, the original self-training loss is $\frac{1}{m+n}\sum_{i=m+1}^{m+n} \ell_\theta(X_i, \hat f(X_i))$. Thus our method is fundamentally different from the original self-training loss whenever $\alpha_t>0$.
>
>
> ## Comment 2
>
> **Reviewer:**
>
>
> >  Image classification experiments were conducted using ImageNet-100. However, when the labeled set ratio is 100%, the top-1 accuracy of DaVIT and ResNet50 is remarkably lower at 47.8% and 46.7%, respectively, compared to the reported top-1 accuracy in other papers. Following the paper of DaVit and ResNet, the top-1 accuracy on the more complex task of ImageNet1k, compared to ImageNet-100, using DaVit-tiny and ResNet is 82.8% and 79.26%, respectively. Following [1], the top-1 accuracy of various resnet50-based methods for ImageNet-100 consistently surpasses 70%. It appears that the baseline has not been sufficiently well-trained.
>
>
>
> **Response:**
>
> - Thank you for your suggestions. **We have included experiments for sufficiently well-trained ImageNet-100 results (training for full 300 epochs) in Figure 1 of the uploaded one-page PDF, and well-trained CIFAR-10-4K, CIFAR-100-10K in Table 1 of the PDF / General Response.** Even in these cases, our method still gives a universal improvement over the baselines.
>
> - Our method gives better improvement when the teacher model is not extremely accurate. If the teacher model is always correct, then our method reduces to the original self-training procedure. This is why previously we focus on the case when the baseline is not sufficiently well-trained. We will add more discussions and all the comparisons including well-trained case and less-trained case in the revision.
>
>
>
> ## Comment 3
>
> **Reviewer:**
>
>
> > There is a lack of comparison with other semi-supervised methods. While this method compares with the basic self-training loss, it is necessary to compare it with various methods that utilize pseudo-labels for self-training [2], [3], [4]. Most semi-supervised image classification methods have performed experiments on benchmark datasets such as ImageNet1k and CIFAR100. These methods achieve a top-1 accuracy of 72% or higher when using a labeled dataset of 10% on ImageNet1k. Therefore, it is necessary to compare the effectiveness of this method on these benchmark datasets. Similarly, in semi-supervised 3D object detection, it is necessary to compare the results with existing baseline methods [5], [6].
>
>
>
> **Response:**
>
> Thank you for your suggestion! **We have included comparisons with other 11 SOTA baselines in Table 1 the uploaded new PDF.** Overall, our method still gives the best improvement, and can be combined with existing baselines. We will add the results and more discussions in the revision as well.
>
> ## Comment 4
>
> **Reviewer:**
>
>
> > Although this paper proposes a method for the distribution mismatch case in Section 2.4, estimating the probability values of each sample, p(x) and q(x), for each distribution is not easy. Practical methods for this issue have not been provided, and experiments on this aspect are also lacking. Practical algorithms and experiments need to be provided in order to address this concern.
>
> **Response:**
>
> Thank you for your comments! We agree that estimating the importance ratio can be hard for practical scenarios when we don't have access to the marginal distributions. We are happy to provide simulated results for this case. However, we would also like to point out that most of the self-training pipeline is based on the setting when there is no distribution mismatch between the labeled and unlabeled samples. Our proposed method makes a first attempt towards provable method to address the distribution mismatch phenomenon, and explains the name `doubly-robust' in statistics. However, we will mark that the proposed method for distribution shift may not be practical due to the difficulty in estimating the importance ratio, which is left as an open problem for future research.
>
> \
> We wish that our response has addressed your concerns, and turns your assessment to the positive side. If you have any questions, please feel free to let us know during the rebuttal window. We appreciate your suggestions and comments! Thank you!

---

### Official Review · Reviewer_bmAm · 2023-07-03

**Soundness:** 3 good
**Presentation:** 3 good
**Contribution:** 3 good
**Rating:** 6
**Confidence:** 3

**Summary:**

The paper proposed a doubly robust loss for self-training. The proposed loss is analysed and shown to have preferable theoretical properties.

**Strengths:**

1. The idea is interesting: a simple change from 1/(m+n) to 1/n (line 51 - 53) lead to a doubly robust loss function for self-training.
2. The writing is clear and easy to follow

**Weaknesses:**

1. While the proposed doubly robust loss for self-training enjoys theoretical advantages, directly minimizing the loss during network training leads to instability. The actual loss used in line 219 is very different especially in the early epochs.

2. Lack of comparison to other stronger semi-supervised learning baselines. For example, the authors discuss MixMatch and FixMatch as related work but did not compare with them in experiments.

minor:
1. I don't think the description in line 79-87 is precise. MixMatch/FixMatch does not pre-trained a teacher model on 'labeled' data.

**Questions:**

NA

---

> ### Author Rebuttal · Authors · 2023-08-06
>
> Thank you for your valuable comments and suggestions. We have corrected all the typos suggested. Please find our responses to each comment below.
>
> ## Comment 1
>
> **Reviewer:**
>
>
> > While the proposed doubly robust loss for self-training enjoys theoretical advantages, directly minimizing the loss during network training leads to instability. The actual loss used in line 219 is very different especially in the early epochs.
>
> **Response:**
>
> Thank you for your comments! We would like to mention that the actual loss is an interpolation between two losses: in the early epochs, it is close to the original pseudo-labeling method, which first uses the pseudo-labels to learn a student model that is similar to the teacher model. In the later epochs, it utilizes the new proposed loss function to correct the learned student model. We find that it stabilizes the training greatly.
>
>
> ## Comment 2
>
> **Reviewer:**
>
>
> > Lack of comparison to other stronger semi-supervised learning baselines. For example, the authors discuss MixMatch and FixMatch as related work but did not compare with them in experiments.
>
>
>
> **Response:**
>
> Thank you for the comments! We have included comparisons with other baselines in **the uploaded new PDF / General Response**, which compares against MixMatch, FixMatch, and another 9 baselines on CIFAR-10, CIFAR-100 datasets. We will add it in the revision as well.
>
>
>
> ## Comment 3
>
> **Reviewer:**
>
>
> > I don't think the description in lines 79-87 is precise. MixMatch/FixMatch does not pre-trained a teacher model on 'labeled' data.
>
>
>
> **Response:**
>
> Thank you for your suggestion! We have corrected our paper to make this part precise. MixMatch / FixMatch are methods that do not rely on pre-training a teacher model on the labeled dataset.
>
> \
> Thanks again for your time and effort! For any other questions, please feel free to let us know during the rebuttal window.

---

> > ### Comment · Reviewer_bmAm · 2023-08-14
> >
> > The authors' rebuttal partly address my concerns.
> > On the one hand I appreciate the simplicity and favorable theoretical properties of the proposed method, but on the other hand, the additional experiments shows that the performance improvement seems modest.
> >
> > I keep my score unchanged.

---

> > > ### Author Response · Authors · 2023-08-15
> > > **Thank you for your response!**
> > >
> > > Thank you for your response! We appreciate your time for providing valuable suggestions and comments, which help greatly improve our paper.
> > >
> > > To add one additional note, our additional experiments mostly work with the case when the teacher model is close to 90% or higher accuracy. In this case, it is expected that the gain for a better teacher model is smaller compared to the gain for a worse teacher model. When the teacher model is perfect (100% accurate), our method reduces to the pseudo-labeling method. And the gain will be 0 compared to naive pseudo-labeling. Thus when the teacher model is close to perfect, there will only be marginal improvement compared with the pseudo-labeling method. However, if we already have a very accurate teacher model, the necessity of re-training a new student model is also unclear.
> > >
> > > Our method really shines when it is uncertain how good the teacher model is, or when the teacher model is not a perfect predictor. This is reflected in our original experiments in the paper. And we can show that even when the teacher model is very accurate, our proposed method can still achieve SOTA performance among all 12 estimators considered.

---

### Official Review · Reviewer_AyWm · 2023-07-05

**Soundness:** 3 good
**Presentation:** 4 excellent
**Contribution:** 3 good
**Rating:** 7
**Confidence:** 5

**Summary:**

The authors propose a very simple yet effective modification to the original loss for self-training by re-weighting terms of the loss function making. This change effectively balances between using the pseudo-labels when the predictor is strong and learning to not use it when it is unreliable, making it doubly robust. They provide a sound theoretical analysis and  empirical evaluation on classification and object detection tasks substantiating their claims.

**Strengths:**

- Strengths
    - The paper is well written and easy to follow with sufficient background and motivated examples given to present the chain of reasoning well.
    - For linear predictor, the proposed loss is unbiased with lower variance which is strictly better than self-training
    - The results on both classification and 3D object detection highlight clear improvement over standard self distillation
    - The technical novelty in the paper is limited with just a small modification to the overall loss. But the theoretical insights including guarantees for general loss and positive experimental results make it a meaningful contribution. The simplicity of the modification also make it much more likely to be adopted and have higher impact.

**Weaknesses:**

- Weaknesses
    - The analysis provided is for very simplistic settings of linear predictor or mean-predictions and it’s unclear how much of it translates to realistic settings of over-parameterized deep nets trained on SGD.
    - For image classification, the baselines used in the paper are meaningful but thorough comparison with other state of the art self-imitation methods like noisy-student etc is lacking.
    - The evaluation is restricted to lower data regimes as the model shines when the training data is limited, making their evaluation a bit more contrived as compared to real world scenarios. It seems the proposed loss should shine more when unlabelled data scales, so some experimentation on the impact on performance as the unlabelled data scales would also be interesting to see
    - One important use of self-distillation is using unlabelled data to show domain adaptation to that domain. I would like to see experiments for that as well.

**Questions:**

* In Section 2.1, the intuitive interpretation for when the predictor is bad is not super clear and paper would benefit from some elaboration. If just m goes to infinity, how would the two terms cancel out

* In the case of distribution mismatch, what’s the motivation for changing the first term to average of (n) sample instead of (m+n)? Also how are $\pi (x)$ ie importance weights chosen?

* Did the authors explore tuning the predictor $\hat{f}$ on the labelled examples before generating the pseudo-examples to train the student?

Typos and possible errors
- Inconsistent notation : line 155 should be $\hat{\theta}$ instead of $\theta^{*}$
- In line 161, I believe the upper bound should be $6/n(var[Y] ..) $ instead of $4/n$. Please verify!
- in equation in line 171, it should be $\theta_{SL}$ instead of $\theta_{DR}$

**Limitations:**

The authors have adequately addressed some limitations. Other suggestions are listed above.

---

> ### Author Rebuttal · Authors · 2023-08-06
>
> Thank you for your valuable comments and suggestions. We have corrected all the typos suggested. Please find our responses to each comment below.
>
> ## Comment 1
>
> **Reviewer:**
>
>
> > The analysis provided is for very simplistic settings of linear predictor or mean-predictions and it’s unclear how much of it translates to realistic settings of over-parameterized deep nets trained on SGD.
>
> **Response:**
>
> Thank you for your comments!  We would like to clarify that we have a **guarantee for arbitrary loss (including deep neural network) in Theorem 2 in Section 2.3**. This shows that even in the case of deep learning, our proposed algorithm still converges to a good point. In contrast, the existing method fails even in the simplest case of mean estimation. Our section 2.2 on mean estimation is only a motivating example for the general case.
>
>
>
> ## Comment 2
>
> **Reviewer:**
>
>
> >  For image classification, the baselines used in the paper are meaningful but thorough comparison with other state-of-the-art self-imitation methods like noisy-student etc is lacking.
>
>
>
> **Response:**
>
> Thank you for the comments! We have included comparisons with other baselines in the uploaded new PDF, which compares against another 12 baseline algorithms in CIFAR-10 and CIFAR-100 dataset. We will add more comprehensive comparisons in the revision as well.
>
>
>
> ## Comment 3
>
> **Reviewer:**
>
>
> > One important use of self-distillation is using unlabelled data to show domain adaptation to that domain. I would like to see experiments for that as well.
>
>
> **Response:**
>
> Thank you for your comments! We have included theoretical results on the right algorithm for distribution mismatch. However, the proposed method requires knowledge about the importance ratio. Making the proposed algorithm practically implementable still remains an open problem.
>
> ## Comment 4
>
> **Reviewer:**
>
>
> > In Section 2.1, the intuitive interpretation for when the predictor is bad is not super clear and paper would benefit from some elaboration. If just $m$ goes to infinity, how would the two terms cancel out?
>
>
>
> **Response:**
>
> Thank you for your comments! The finite-sample guarantee when none of $m, n$ goes to infinity is given in Section 2.2 for mean estimation, and Section 2.3 for general loss functions. In short, if only $m$ goes to infinity, there will an additional noise introduced whose standard deviation is proportional to $1/\sqrt{n}$. This can be seen from the third equation in Proposition 1, or Theorem 2.
>
>
> ## Comment 5
>
> **Reviewer:**
>
>
> >  In the case of distribution mismatch, what’s the motivation for changing the first term to average of $(n)$ sample instead of $(m+n)$? Also how are the importance weights chosen?
>
>
> **Response:**
>
> Thank you for your comments! Sorry here is a minor typo. We would like to change the first term from the average of $(m+n)$ samples to the average of $m$ samples. This is due to that the marginal distribution of the first $m$ (unlabeled) samples is different from the distribution of the last $n$ (labeled) samples. If we mix both distributions together, the resulting distribution won't be the same as the distribution of the unlabeled samples. And in this case, we cannot use let the first two terms cancel each other due to distribution mismatch. Thus in the first term, we only take average with the $m$ unlabeled samples. In practice, the importance weights are required to be known or estimated from the representations of the features $X$.
>
> ## Comment 6
>
> **Reviewer:**
>
>
> >  Did the authors explore tuning the predictor on the labelled examples before generating the pseudo-examples to train the student?
>
>
>
> **Response:**
>
> Thank you for your comments! Yes, we select the best checkpoint for the predictors on the labeled examples. And then generate the pseudo-examples to train the students. This applies to both the baseline algorithms and our proposed algorithm.
>
> \
> Thanks again for your time and effort! For any other questions, please feel free to let us know during the rebuttal window.

---

> > ### Comment · Reviewer_AyWm · 2023-08-16
> >
> >
> > Thank you, the authors have addressed some concerns and weaknesses. The newly added results also look promising. I would like increase the rating to accept.

---

### Official Review · Reviewer_yYuG · 2023-07-07

**Soundness:** 2 fair
**Presentation:** 2 fair
**Contribution:** 2 fair
**Rating:** 5
**Confidence:** 4

**Summary:**

This paper proposes a pseudo-labeling approach that balances out the supervised signal between the labeled and incorrect pseudo-labeled datapoints during the training process. The aim is to only account for the pseudo-labels when they are correctly labeled, which may happen when the covariate distribution of the unlabeled dataset and the labeled dataset matches. They show some analysis of the proposed loss and results in ImageNet100 (a subset with 100 random classes from ImageNet-1k) and mini-ImageNet100  and nuScenes dataset.

**Strengths:**

The proposed method is clearly written, and the paper can easily be understood.
The paper with all necessary details and deep explanations for experimental results.
Comprehensive algorithmic analysis and clear motivation.
The proposed method effectively improves over vanilla pseudo-labeling [1]

[1] D.-H. Lee. Pseudo-label : The simple and efficient semi-supervised learning method for deep neural networks. ICML 2013 Workshop : Challenges in Representation Learning (WREPL), 07 2013.

**Weaknesses:**

Novelty and missing prior work: Pseudo-labeling is the defacto method for entropy regularization techniques in semi-supervised learning problems. This is a paper that tackles the semi-supervised learning problem. Recent work has explored different ways to mitigate the error propagation from the teacher model and confirmation bias present in pseudo-labeling approaches. For example, [2,3,4] investigate the thresholding effect via fixed and curriculum based approaches, with flexible thresholds that are dynamically adjusted for each class according to the current learning status. With such prior exploration and no comparison with any of that work, it's difficult to assess the importance and impact of this work, which seems limited and incomplete.

The paper in its current state also fails to provide technical details to validate fair comparisons in the main text. Furthermore, no ablations for any of the technical selections are conducted.
The alternative loss proposed for distribution mismatch (Section 2.4) is only shown in the method but not in the empirical section.

[2] Eric Arazo, Diego Ortego, Paul Albert, Noel E O’Connor, and Kevin McGuinness. Pseudolabeling and confirmation bias in deep semi-supervised learning. In IJCNN, pages 1–8, 2020.

[3] Paola Cascante-Bonilla, Fuwen Tan, Yanjun Qi, and Vicente Ordonez. Curriculum labeling: Revisiting pseudo-labeling for semi-supervised learning. In Proceedings of the AAAI Conference on Artificial Intelligence, volume 35, pages 6912–6920, 2021.

[4] Zhang B, Wang Y, Hou W, Wu H, Wang J, Okumura M, Shinozaki T. Flexmatch: Boosting semi-supervised learning with curriculum pseudo labeling. Advances in Neural Information Processing Systems. 2021 Dec 6;34:18408-19.

**Questions:**

Prior literature [2,3,4,5] has shown that predefined threshold values impact the overall performance in pseudo-labeling. Using 0.3 as the threshold seems to be too permissive, allowing too many noisy pseudo-labels. How sensitive is this threshold in your setup?

[5] Oliver A, Odena A, Raffel CA, Cubuk ED, Goodfellow I. Realistic evaluation of deep semi-supervised learning algorithms. Advances in neural information processing systems. 2018;31.

**Limitations:**

No limitation section is provided.

---

> ### Author Rebuttal · Authors · 2023-08-06
>
>
> Thank you for your valuable comments and suggestions. Please find our responses to each comment below.
>
> ## Comment 1
>
> **Reviewer:**
>
>
> > Novelty and missing prior work: Pseudo-labeling is the defacto method for entropy regularization techniques in semi-supervised learning problems. This is a paper that tackles the semi-supervised learning problem. Recent work has explored different ways to mitigate the error propagation from the teacher model and confirmation bias present in pseudo-labeling approaches. For example, [2,3,4] investigate the thresholding effect via fixed and curriculum based approaches, with flexible thresholds that are dynamically adjusted for each class according to the current learning status. With such prior exploration and no comparison with any of that work, it's difficult to assess the importance and impact of this work, which seems limited and incomplete.
>
> **Response:**
>
> Thank you for your comments!
>
> - We are happy to include comparisons with existing methods [2-4]. Please find the **Table 1 in the uploaded PDF / General Response** as our comparisons with other existing baselines. Due to the long-running time for all methods, for now we only include the comparisons with [2]. We also compare with other existing 11 benchmark algorithms for image classification in Table 1 of the uploaded PDF. One can see that our method still achieves better performance than existing methods.
>
> - Our proposed methodology is **fundamentally different from the existing pseudo-labeling idea in [2-4]**. In [2-4], the effect of thresholding is extensively studied. However, when the teacher model is very inaccurate, one can never expect the confidence of each label is given correctly, and thus the thresholding would not provide gain for this case. In fact, we can show that even in the simplest case of mean estimation, all the methods in [2-4] still fail to provide right solution even with infinite number of samples.
>
> - As an orthogonal approach, we propose a simple loss function that automatically uses the labeled samples to test the validity of the teacher model. It is guaranteed to perfectly interpolate both cases: when the teacher model is completely wrong, we will only use the labeled samples; when the teacher model is completely correct, it will use all pseudo-labels. This is not achievable by the existing algorithms in [2-4] since the confidence on each label might be inaccurate as well.
>
> - Our proposed method is **always unbiased and guaranteed to converge to the right solution, in contrast to any existing methods**. We show theoretically that the existing pseudo-labeling idea will give very poor performance when the teacher model is inaccurate. And only the proposed doubly-robust estimator will remain unbiased even in the simplest case such as mean estimation. Our experimental results validate our theoretical predictions.
>
> - As we also show in the experiments for 3D object detection, our method can be directly combined with any threshold method in [2-4] and improve over the existing threshold-based method. We believe the proposed method is a very important and necessary alternative / add-on for all the existing pseudo-labeling ideas.
>
>
>
> ## Comment 2
>
> **Reviewer:**
>
>
> >  The paper in its current state also fails to provide technical details to validate fair comparisons in the main text. Furthermore, no ablations for any of the technical selections are conducted. The alternative loss proposed for distribution mismatch (Section 2.4) is only shown in the method but not in the empirical section.
>
>
>
> **Response:**
>
> Thank you for the comments!
>
> - We will add the details about all the hyperparameters in the revised draft.
>
> - We **have ablation studies for both the curriculum settings and the number of epochs** in Appendix B. We have also added the new ablation study in Table 3 for the pseudo-labeling threshold in the uploaded new PDF file, we will add more data points and ablations on the original pseudo-labeling methods in the final revision.
>
> - The alternative loss for distribution mismatch requires the knowledge of the importance ratio between the target distribution and the original data distribution. We are happy to include simulated results. However, estimating such importance ratio in practical scenarios can be hard.
>
>
>
> ## Comment 3
>
> **Reviewer:**
>
>
> > Prior literature [2,3,4,5] has shown that predefined threshold values impact the overall performance in pseudo-labeling. Using 0.3 as the threshold seems to be too permissive, allowing too many noisy pseudo-labels. How sensitive is this threshold in your setup?
>
>
> **Response:**
>
> Thank you for your comments! We have added the new ablation study for the pseudo-labeling threshold in **Table 3 of the uploaded rebuttal PDF / General Response**. One can see that if we improve the threshold from 0.3 to 0.5 to allow less noisy pseudo-labels, the performance indeed downgrades. We will include more data points and the ablations for the original pseudo-labeling method in the final revision.
>
> \
> We wish that our response has addressed your concerns, and turns your assessment to the positive side. If you have any questions, please feel free to let us know during the rebuttal window. We appreciate your suggestions and comments! Thank you!

---

> > ### Comment · Reviewer_yYuG · 2023-08-21
> >
> > Thanks for the detailed explanation and additional experimental results, I've also read the thoughtful discussion between other reviewers and the authors that helped me clarify some of my questions. However, experimental results show marginal improvements, along with important baselines missing.
> > It is also not clear what the authors mean by: "we propose a simple loss function that automatically uses the labeled samples to test the validity of the teacher model" -- pseudo-labeling methods do the same at each iteration, and the teacher model is validated using a validation set, which contains the true labels; thus, it is hard to say the proposed method does something different from traditional pseudo-labeling and it's variations.
> > In addition, pseudo-labeling approaches are really fast to train, and the datasets are very small. Unfortunately, table one only shows comparisons against consistency regularization methods. Given the nature of the proposed approach, it is important to make fair comparisons with existing entropy regularization methods. Thus, I updated my score to borderline accept.

---

> > > ### Author Response · Authors · 2023-08-21
> > >
> > > Thank you for your response! Below is a clarification to our previous message "we propose a simple loss function that automatically uses the labeled samples to test the validity of the teacher model" and its difference with traditional pseudo-labeling method:
> > >
> > > Our main message is the sentence after the quoted sentence: such simple doubly-robust loss leads to a guaranteed unbiased estimation in the case of both mean estimation and general neural networks. In contrast, even though the teacher model is validated using a validation set, training with traditional pseudo-labeling based methods is still biased. We agree that in some algorithms, pseudo-labeling methods also validate and filter the unlabeled samples based on the ground truth labels. So we will make it precise in our revision and add more clarifications on the main differences between our algorithms and traditional pseudo-labeling methods.
> > >
> > > Thank you again for your comments and suggestions!

---

### Author Rebuttal · Authors · 2023-08-09

We would like to thank the reviewers for the valuable comments and suggestions, which help us greatly improve our paper. Besides individual responses, we summarize the revision and new experimental results in the one-page PDF uploaded. We also include the markdown table for your reference, which are the same results as the PDF.

### Additional Experiment 1 (Table 1): Comparisons with previous SOTAs on CIFAR-10 and CiFAR-100

In Table 1 of the uploaded PDF, we compare with **another 11 baselines** in terms of error rate on CIFAR-10-4K and CIFAR-100-10K under the same settings (i.e., Wide ResNet-28-2 for CIFAR-10 and WRN-28-8 for CIFAR-100). We show that our method is only 0.04 inferior to the best method Meta Pseudo Labels for CIFAR-10-4K, and achieves the best performance for CIFAR-100-10K. (The numbers are not reported in some of the methods in CIFAR-100-10K, we will try to re-implement and re-run these methods in the future.)

| Method     | CIFAR-10-4K (error rate, \%) | CIFAR-100-10K (error rate, \%) |
| ------------------ | ------------------- | --------------------- |
| Pseudo-Labeling    | 16.09               | 36.21                 |
| LGA + VAT          | 12.06               | --                    |
| Mean Teacher       | 9.19                | 35.83                 |
| ICT                | 7.66                | --                    |
| SWA                | 5.00                | 28.80                 |
| MixMatch           | 4.95                | 25.88                 |
| ReMixMatch         | 4.72                | 23.03                 |
| EnAET              | 5.35            | --                    |
| UDA                | 4.32                | 24.50                 |
| FixMatch           | 4.31                | 23.18                 |
| Meta Pesudo Labels | **3.89**            | --                    |
| **Ours**           | 3.93                | **22.30**             |
### Additional Experiment 2 (Figure 1): Sufficiently well-trained ImageNet-100

In our original experiments, we mostly focus on a teacher model that is not super accurate, since our method reduces to the original pseudo-labeling when the teacher model is completely correct for all labels. In this experiment, we fully train the teacher model with 300 epochs on ImageNet-100, leading to the accuracy of the teacher model as high as 88.4%. We show that even in this case, our method outperforms the original pseudo-labeling baseline.

|  Data Fraction    |   Labeled Only  (acc, \%) |   Pseudo + Labeled (acc, \%)  |   Ours (acc, \%)  |
| ---- | ---- | ---- | ---- |
|   20   | 63.59 | 64.87 | **67.16** |
|   50   | 81.23 | 82.92 | **85.47** |
|   80   | 85.50 | 86.98 | **87.57** |
|   100   | 88.01 | 89.43 | **90.61** |

### Additional Experiment 3 (Table 2): Comparisons with previous SOTAs on nuScenes object detection dataset

We compare with the idea of pseudo-labeling + confirmation bias in [1]. And show that on the object detection dataset, our method still gives better performance for various labeled data fractions. Due to the time limit, we are still not yet finished all the experiments with varying labeled fractions and other methods. We will include more comparisons in the final revision.

| Labeled Fraction | Labeled Only (mAP↑) | Labeled Only (NDS↑) | Labeled + Pseudo (mAP↑) | Labeled + Pseudo (NDS↑) | Doubly robust Loss (mAP↑) | Doubly robust Loss (NDS↑) | Pseudo-Labeling + Confirmation Bias (mAP↑) | Pseudo-Labeling + Confirmation Bias (NDS↑) |
|:----------------:|:-------------------:|:-------------------:|:-----------------------:|:-----------------------:|:-------------------------:|:-------------------------:|:-------------------------------------------:|:-------------------------------------------:|
| 1/24       | 7.56        | 18.01       | 7.60        | 17.32         | **8.18**       | **18.33**          | 7.80      | 16.86      |
| 1/16       | 11.15      | 20.55     | 11.60      | 21.03         | **12.30**      | 22.10           | 12.15        | **22.89**     |


### Additional Experiment 4 (Table 3): Ablation on the detection thresholds

We include ablation studies for different detection thresholds, showing that our choice of 0.3 is not too pessimistic. We will include more data points and the results on the original pseudo-labeling methods in the final revision.
| Labeled Data Fraction | τ = 0.3 (mAP↑) | τ = 0.3 (NDS↑) | τ = 0.5 (mAP↑) | τ = 0.5 (NDS↑) |
|:----------------------:|:--------------:|:--------------:|:--------------:|:--------------:|
| 1/24                   | **8.18**       | **18.33**      | 4.37           | 13.17          |
| 1/16                   | **12.30**      | **22.10**      | 8.09           | 19.70          |

### Novelty and Contributions

We would like to remark here that our proposed method is fundamentally different from **all existing pseudo-labeling-based methods.**
- Our proposed method is the first to have convergence guarantee for any loss functions, including neural networks. (see Theorem 2). While most of the SOTA methods are built up on the original pseudo-labeling idea, which is biased even for the simplest setting of mean estimation.
- Based on the experimental results, our proposed method improves over almost all existing baselines, showing the power of the doubly-robust idea.
- Our method is the first that connects and unifies the idea of doubly robust estimator in statistics for propensity score estimation, the prediction-powered inference idea for confidence estimation, and the self-training idea in knowledge distillation and neural network training.
- Our method proposes an alternative loss that uses the labeled data to `correct' the unlabeled samples. It can also be combined with any of the existing pseudo-labeling methods to further improve their performance.

We hope that our pointwise responses below could clarify all reviewers’ confusion. We thank all reviewers’ time again and we are always ready to solve your concerns.

---

> ### Author Response · Authors · 2023-08-19
> **Finished Additional Experiment 4**
>
> Dear reviewers,
>
> We have finished the additional experiments 4 on the ablation studies on the detection thresholds. As is shown in the table, setting $\tau = 0.5$ does not improve compared with the case of $\tau=0.3$ in our default choice. And our doubly robust estimator still remains the best estimator under the most appropriate choice of $\tau$. We will supply more experiments in the final version. Thank you again for the great comments! Please let us know if you have any further suggestions.
>
> | Labeled Data Fraction | $\tau = 0.3$ | | | | $\tau = 0.5$ | | | |
> |:----------------------|:-------------:|:-:|:-:|:-:|:-------------:|:-:|:-:|:-:|
> |                       | Labeled+Pseudo | | Doubly Robust Loss | | Labeled+Pseudo | | Doubly Robust Loss |
> |                       | mAP↑ | NDS↑ | mAP↑ | NDS↑ | mAP↑ | NDS↑ | mAP↑ | NDS↑ |
> | 1/24                   | 7.56 | 18.01 | **8.18** | **18.33** | 7.15 | 15.82 | 4.37 | 13.17 |
> | 1/16                   | 11.15 | 20.55 | **12.30** | **22.10** | 11.05 | 21.22 | 8.09 | 19.70 |

---

### Decision · Program_Chairs · 2023-09-21

**Decision:**

Accept (poster)

**Comment:**

The paper received borderline to positive reviews. The paper proposes a novel methodology for self-supervised learning where the method has some robustness to inaccurate pseudo-labels. The paper provides both empirical and theoretical justifications that most reviewers found compelling. Suggestions were made to discuss and compare to additional methods and the authors have partially addressed this in the rebuttal. In general the paper seems to make some reasonable contribution to address a current weakness in self-supervised learning and the method demonstrates positive results. The paper is recommended for acceptance.